

# Dynamics of the electrojet during intense magnetic disturbances

Liudmila I. Gromova[1], Matthias Förster[2,3], Iakov I. Feldstein[1], and Patricia Ritter[2]

[1]Institute of Terrestrial Magnetism, Ionosphere, and Radiowave Propagation of the Russian Academy of Sciences (IZMIRAN), 142090 Troitsk, Moscow region, Russia

[2]Helmholtz-Centre Potsdam, GFZ German Research Centre for Geosciences, 14473 Potsdam, Germany

[3]Max-Planck-Institut für Sonnensystemforschung, 37077 Göttingen, Germany

*Correspondence to:* M. Förster
mfo@gfz-potsdam.de

**Abstract.** Hall current variations in different time sectors during six magnetic storms of the summer seasons in 2003 and 2005 are examined in detail: three storms in the day-night meridional sector and three storms in the dawn-dusk sector. We investigate the sequence of the phenomena, their structure, positions and the density of the polar (PE) and the auroral (AE) Hall electrojets using scalar magnetic field measurements obtained from the CHAMP satellite in accordance with the study of Ritter et al. (2004a). Particular attention is devoted to the spatial-temporal behaviour of the PE at ionospheric altitudes during daytime hours both under geomagnetically quiet and under magnetic storm conditions. We analyze the correlations of the PE and AE with various activity indices like SYM/H and ASYM/H, that stand for large-scale current systems in the magnetosphere, AL for ionospheric currents, and the IndN coupling function for the state of the solar wind. We obtain regression relations of the magnetic latitude MLat and the electrojet current density $I$ with those indices and with the interplanetary By and Bz magnetic field components. For the geomagnetic storms during summer seasons investigated here, we obtain the following typical characteristics for the electrojets' dynamics:

1. The PE appears at magnetic latitudes (MLat) and local times (MLT) of the cusp position.

2. This occurs in the daytime sector at MLat~73°-80° with a westward or an eastward direction, depending on the orientation of the IMF By component. Changes of current flow direction in the PE can occur repeatedly during the storm, but only due to changes of the IMF By orientation.

3. The current density in the PE increases with the intensity of the IMF By component from





I∼0.4 A/m for By∼0 nT up to I∼1.0 A/m for By∼23 nT.

4. The MLat position of the PE does not depend on the orientation and the strength of the IMF By component. It depends, however, on the strength of the IMF Bz component.

5. The PE is situated at MLat∼73° on the dayside during geomagnetically quiet periods and the recovery phase of a magnetic storm, and it shifts equatorward during intense substorms and the main phase of a storm.

6. There is no connection between MLat and the current density $I$ in the PE with the magnetospheric ring current DR (index SYM/H).

7. There is a correlation between the current density $I$ in the PE and the partial ring current in the magnetosphere (PRC, index ASYM/H), but practically no correlation of this index with MLat of the PE.

8. Substorms that occur before and during the beginning of a storm main phase are acompanied in the daytime by the appearance of an eastward electrojet (EE) at MLat∼64°, and then also by a westward electrojet (WE). In the nighttime sector the WE appears at MLat∼64°.

9. During the development of the main storm phase, the daytime EE and the nighttime WE shift toward subauroral latitudes of MLat∼56° and intensify up to I∼1.5 A/m. Both electrojets persist during the main phase of the storm. The WE is then located about 6° closer to the pole than the EE during evening hours and about 2°-3° during daytime hours.

**Keywords.** high-latitude magnetic variations, magnetic storm phases, interplanetary magnetic field, Hall current modelling, equivalent ionospheric currents, current systems in the magnetosphere, polar electrojet, eastward and westward auroral electrojets, SYM/H, ASYM/H, AL, IndN

## 1 Introduction

An intense study of the polar electrojet (PE) at the high-latitude daytime ionosphere was initiated by the works of Svalgaard (1968) and Mansurov (1969). They had demonstrated that its characteristic magnetic field variation depends on the sector structure of the interplanetary magnetic field (IMF) that is much alike the average magnetic field of the solar photosphere. The IMF lines up in a spiral structure near the ecliptic plane with IMF Bx>0 and By<0 (toward the sun) or Bx<0 and By>0 (away from the sun), where one prevalent direction is kept usually for several days. During summer the sector structure is accompanied by intense variations of the geomagnetic Z component within the polar cap at $\Phi \sim 86°$ and of the H component at $\Phi \sim 78°$. An increase or decrease of the magnetic field components with respect to the quiet time is determined by the activity level and the IMF sector





structure. $\Delta Z < 0$ and $\Delta H > 0$ is found for the away sector, and the opposite variations for the toward sector.

The IMF sector structure does not always correspond to the expected magnetic field variations in the near-polar region. Friis-Christensen et al. (1972) showed that during periods of discrepancy between the expected magnetic variations and the sector structure from satellite observations, there existed always an essential deviation of the IMF from the usual spiral structure. During these cases, the azimuthal IMF By component was oppositely directed to the expected direction of the spiral. This implies, that the magnetic variation on ground is not primarily controlled by the sector structure (toward or away the sun), but by the azimuthal component of the IMF (eastward or westward).

Various methods have been developed for the extraction of the PE magnetic field variations from groundbased observations in the near-polar region (Feldstein, 1976). The most effective approach appeared to be the correlation method (Jørgensen et al., 1972; Friis-Christensen and Wilhjelm, 1975; Feldstein et al., 1975b). It is based on the fact, that both the direction of the PE and its intensity depend on the IMF By component. The method allows to separate the magnetic variations of the PE from variations of other sources and to show the spatial-temporal variation of the PE vector variations very clearly. Feldstein et al. (1975b) described the findings of a geomagnetically quiet interval in summer 1965 and the characteristics of the equivalent current system, controlled by the IMF By component. In a first step, time intervals with correlations of the magnetic X(H), Y, and Z components with IMF By were identified for observatories with $\Phi > 65°$. In case of existing correlations, they appeared to be practically always close to a linear dependence with a correlation coefficient r. Correlation was assumed to exist for values of $r > 0.4$; otherwise (for $r \leq 0.4$) it was assumed as non existing. Such a boundary for significant r values is justified by the correlation correction $S_r = (1 - r^2)/\sqrt{(n - 1)}$. For values of $|r/S_r| \geq 3$, the relation between the X(H), Y, and Z components with the IMF By cannot be regarded as accidental. With $n \sim 50$ the correlation is not randomly distributed for $r > 0.4$.

Regression lines, which relate the ground magnetic variations with the IMF By component, were estimated for all MLT, based on the observed intervals with $r > 0.4$. They were used to describe the spatial-temporal distribution of the magnetic variations in the horizontal and vertical plane, and finally for the estimation of the equivalent current system for IMF By = 6nT. Its integral intensity amounts to 180 kA with a maximum current density of the electrojet in the dayside sector of $\sim 0.5 A/m$ at $80° < \Phi < 81°$. An analogous estimation for July-August 1966 resulted in a value of $\sim 0.35 A/m$ at the same latitudes (Sumaruk and Feldstein, 1973).

The PE in the dayside sector does not disappear during magnetic disturbances (Feldstein et al., 2006). The PE shifts equatorward to $72° < \Phi < 74°$ in the longitudinal range of 08<MLT<17 during intense substorms (with AL$\sim$-800 nT), and during periods of geomagnetic storms with AL$\sim$-1200 nT and Dst$\sim$-150 nT it is situated at $66° < \Phi < 68°$ between 09<MLT<15. The current intensities of the PE increases only slightly to about $\sim$0.5 A/m.





The variations of the magnetic field at the Earth's surface at high latitudes, which were derived

with the method of regression analysis, allowed to determine the IMF By control of the spatial-temporal distributions of the electric field potential at ionospheric altitudes as well as the ionospheric and field-aligned currents (FACs) (Friis-Christensen et al., 1985; Feldstein and Levitin, 1986). The electric field potential for an inhomogeneous ionospheric conductivity is obtained by solving a second-order partial differential equation. Friis-Christensen et al. (1985) used magnetic observa-

tions of the summer seasons in 1972 and 1973, while Feldstein and Levitin (1986) obtained it for summer 1968. The potential differences at cusp latitudes in the daytime sector are ∼20 kV for IMF By∼ ±6 nT.

Leontyev and Lyatsky (1974) postulated a penetration of the solar wind electric field into the magnetosphere at daytime cusp latitudes. This electric field is generated by the potential difference

between the northern and southern boundaries of the magnetotail. Under the assumption of high conductivities along the magnetic field lines, the electric field exists only at open field lines, which have their footprints in the polar caps, and will be short-circuited along closed field lines. The model allowed to estimate the effectivity of the solar wind electric field penetration into the magnetosphere to ∼10%.

Olsen (1996) used MAGSAT magnetic field data in a height range of 350<h<550 km to determine the strength and location of the auroral electrojets at 115 km altitude. He showed for the first time the possibility to estimate the horizontal ionospheric currents from scalar magnetic measurements only. The ionospheric currents were modelled by hundreds of infinite linear currents perpendicular to the orbital plane of the spacecraft with discretisation intervals of 111 km. The problem of iono-

spheric current estimation is underdetermined and its solution is not unique. In order to constrain the solution, a regularization method is used. The compilation of modelled and measured variations of the magnetic field along the satellite orbit on December 04, 1979, 17:00 UT, demonstrated the good agreement for the field-aligned component, but a significant discrepancy for the field-perpendicular one. The discrepancy is mainly caused by magnetic fields of the FACs. The integral amplitude of the

ionospheric currents during the interval November 28 till December 10, 1979, yielded a correlation of $r = 0.88$ with the AE-index.

The IMF By orientation influences not only the PE, but also the movements of the auroral forms at cusp latitudes (Sandholt et al., 2002). Simultaneously with permanently poleward moving discrete auroral forms at the equatorward boundary of the cusp, which are controlled by the IMF Bz

component, there exist east-west moving auroral forms. This azimuthal movement is controlled by IMF By, such that for By>0 the discrete forms move westward and for By<0 eastward. The movement of the auroral forms is in opposite direction to the PE current flow direction. This can be expected, because the discrete auroral forms and the channels of enhanced ionospheric conductivity are both due to precipitating electrons into the upper atmosphere. A detailed consideration of the

interrelation between auroral luminosity, auroral particle precipitation, and the PE during magnetic





disturbances was given by Sandholt et al. (2004). As shown there, the strong convection channel is located on the dawn side of the polar cap for IMF By>0, and on the dusk side for By<0 conditions. The electron precipitation in the regime of the convection channel in the morning sector consists of a band (∼500 km) of structured precipitation. The PE is located on the high-latitude boundary of the

structured luminosity region in the vicinity of the strong flow channel of magnetospheric convection close to the bright auroral arc. For By>0, this channel is located in the morning sector on the polar cap boundary with FAC out of the ionosphere, and FAC into the ionosphere equatorward of the polar cap boundary.

Ritter et al. (2004b) investigated variations in the location and density of the auroral electrojets,

which were independently determined both from ground-based (IMAGE magnetometer network) and satellite (CHAMP) measurements. For the estimation of the Hall current from CHAMP data, a current model consisting of a series of 160 current lines were placed at an altitude of 110 km and separated by 1° in latitude. The magnetic field of the line currents were related to the current strength $I$ according to the Biot-Savart law. The density of each of the 160 line currents were derived from

an inversion of the observed field residuals using a least-square fitting approach. They determined the geomagnetic latitude and current densities of the eastward (EE) and westward electrojets (WE) in the evening, nighttime, and morning sectors.

Two- or one-dimensional ionospheric Hall current systems were independently determined from variations of the horizontal magnetic field, measured by the IMAGE ground-based magnetometer

network. Comparisons of satellite with ground-based measurements of ionospheric currents at auroral latitudes have been done for satellite passages during magnetic storms as, e.g., that of 5–6 November 2001, during substorms, and according to statistical data. The ratio of the current densities from IMAGE and CHAMP was provided for a latitudinal range of 60°-77°as well as mean values of the current densities, variations of the correlation coefficients, and coefficients of the re-

gression equations. The ratio of the current densities and of the correlation coefficient as determined from currents above and below the ionosphere, is close to unity. Such a correspondence between the results of two different model approaches constitute the base for the following statements: a) the estimation of the position and density of the auroral electrojets can be carried out with observations above the ionospheric current layer by means of low-Earth orbiting (LEO) satellites; b) the

currents can be estimated from scalar magnetic field measurements; c) the result of the calculations are the parameters of the Hall current at an altitude of ∼110 km (where the maximum value of the ionospheric Hall conductivity occurs). This method of Hall current estimation from satellites was proposed for the first time by Olsen (1996). Its detailed justification has been validated quantitatively by ground-based observations by Ritter et al. (2004b).

Based on magnetometer data of the IMAGE and EISCAT networks, Feldstein et al. (1997) showed that the electrojets shift equatorward during the main phase of strong magnetic storms. For DST∼ −300 nT, the EE in the evening and the WE in the nighttime and early morning hours shifts to



$\sim 54°$–$\sim 55°$. Feldstein and Galperin (1999) studied the correlation between EE and WE with the structure of plasma precipitations of 30 eV–30 keV according to DMSP F08, F10, and F11

satellite observations during the magnetic storms of 10-11 May 1992, 05-07 February 1994, and 21-22 February 1994. The EE displaces in the region of diffuse aurora, equatorward of the discrete auroral forms, and projects along magnetic field lines into the inner magnetosphere between the plasmasphere and the central plasma sheet of the magnetospheric tail. The WE is located at the auroral oval and projects along magnetic field lines toward the central plasma sheet in the tail.

Wang et al. (2008) made use of the Hall current estimations for the intense magnetic storms of 31 March to 01 April 2001 and 17–21 April 2002 to investigate the position and current densities of auroral electrojets (WE and EE) as well as the relations of the electrojets to the Dst index and the IMF Bz component. The currents were determined from scalar magnetic field measurements of the CHAMP satellite (orbit in the meridional plane of 15–03 MLT and 16–04 MLT) according to the

method that was proposed by Ritter et al. (2004a).

In this study we investigate not only the auroral electrojet, but also the polar electrojet characteristics during six intense magnetic summer storms. In this introduction section we have briefly described the historic progression of method used to determine the ionospheric currents from space observations with LEO satellites and from ground magnetic field data. In section 2 we present an

overview of the CHAMP data used as well as the indices, which characterize the electro-magnetic conditions in the near-Earth space during the geomagnetic storms under study. Section 3 provides a short description of the method for the determination of the Hall currents from CHAMP scalar magnetic records. In section 4 we consider the latitudinal variation of the density and position of the electrojets during different phases of the magnetic storm on 29–30 May 2003. Particular attention

is drawn to the polar electrojet (PE). The discussion of the control of the current direction in the electrojets, its density and latitudinal position by various indices, which characterize the disturbance level and the effectivity of the interaction of the interplanetary medium on the magnetospheric processes follows in the subsections 5.1 and 5.2 for the polar electrojet (PE) and the auroral electrojets (AE), respectively. The Conclusion's section 6 summarizes the main results of the study with respect

to the Hall current variations during the various storm phases (subsection 6.1), the polar electrojet (6.2), and the auroral electrojet within four different time sectors (6.3).

## 2   Data

The CHAllenging Minisatellite Payload (CHAMP) spacecraft (Reigber et al., 2002) was launched on 15 July 2000 into a circular, near-polar orbit with an inclination of 87.3°. From its initial orbital

height at $\sim 460$ km, it has decayed to $\sim 400$ km in 2003 and $\sim 350$ km after 5 years. The orbital plane precesses to earlier local times at a rate of about one hour per 11 days so that the orbit covers all local times within about 131 days. The data used in this study are scalar magnetic field measurements



**Table 1.** Overview of CHAMP satellite orbits used for this study.

| Date & Time (UT, hrs) | CHAMP orbit numbers | MLT range (hrs) | |
| --- | --- | --- | --- |
| | | ascending | decending |
| 29/30 May 2003, 16–10 | 16229–16240 | ∼14–16 | ∼02–04 |
| 24 Aug 2005, 07–20 | 29012–29020 | ∼11–12 | ∼23–24 |
| 18 Jun 2003, 03–18 | 16532–16541 | ∼12–16 | ∼00–04 |
| 30 May 2005, 02–17 | 27658–27667 | ∼19–21 | ∼06–09 |
| 15 May 2005, 00–19 | 27423–27432 | ∼20–22 | ∼08–10 |
| 18 Aug 2003, 00–23 | 17480–17494 | ∼07–09 | ∼19–21 |

obtained with the Overhauser Magnetometer (OVM) at the boom tip with a resolution of 0.1 nT. In order to isolate the magnetic effect of ionospheric currents in the satellite data, the contributions from all other sources have been removed from the scalar field readings as described in the study of Ritter et al. (2004a).

The CHAMP orbital intervals during various storm periods used for this study are listed in Table 1. The quantity, locations, and intensity of the peaks along the latitudinal current density distribution varies over the course of the storm development. For the description of the storm development, we utilise various solar and geomagnetic indices.

First, we employ the auroral electrojet index (AE), which is derived from geomagnetic variations in the horizontal component observed at selected (10-13) observatories along the auroral zone in the Northern Hemisphere (http://wdc.kugi.kyoto-u.ac.jp/aedir/index.html). The lower envelope (AL) of the superposed plots of all the data from these stations as functions of UT is used in this study.

Further, we employ the SYM/H and ASYM/H indices, which describe the geomagnetic disturbances at mid-latitudes in terms of longitudinally asymmetric (ASY) and symmetric (SYM) disturbances for the H component (http://wdc.kugi.kyoto-u.ac.jp/aeasy/index.html or, alternatively, http://omniweb.gsfc.nasa.gov/ow_min.html). SYM-H is essentially the same as the Dst index, but with a different time resolution (1-min cadence).

Finally, Newell et al. (2007) proposed a new solar wind coupling function Index N (IndN) for the correlation analysis in the solar-terrestrial physics:

$$IndN = d\Phi_{MP}/dt = v^{4/3}B_T^{2/3}\sin^{8/3}(\theta_c/2) \tag{1}$$

Here, $v$ describes the solar wind speed or, more precisely, the transport velocity of IMF field lines that approach the magnetopause, $B_T$ is the magnitude of the IMF, and the IMF clock angle $\theta_c$ is defined by $\theta_c = \arctan(B_y/B_z)$. This function describes best the interaction between the solar wind and the magnetosphere over a wide variety of magnetospheric activity. IndN has a strong correlation with other indices that characterize both the plasma and the IMF in the solar wind as well as the





processes in the magnetosphere. By means of a statistical study of the electrojet characteristics, the new function IndN was used together with the classical indices SYM/H, ASYM/H, and AL.

## 3 Method

The Hall current flows at high latitudes are derived from CHAMP scalar magnetometer records along the satellite orbits according to the method that was proposed by Ritter et al. (2004b). These calculations make use of a current model consisting of a series of infinite current stripes at an altitude of 110 km, the magnetic field of which corresponds to the measured values. The model does not take

into account the contributions from field-aligned and Pedersen currents, measured at CHAMP altitudes. The comparison with ground-based geomagnetic variations of the horizontal component that considers only the contributions from the ionospheric Hall current field, because the contributions from the field-aligned and the Pedersen currents cancel there each other, showed the applicability with high reliability of the modelling assumptions by Ritter et al. (2004b) even for the estimation of

the Hall currents.

The density of ionisation in the near noon hours at latitudes of $75° < \Phi < 80°$ decreases from summer to winter season by about an order of magnitude (Feldstein et al., 1975a). The PE current density amounts during winter to $\sim0.1$ A/m, which makes it difficult to be measured adequately by magnetometers onboard of satellites. Because of that we investigate in this study summer storms

only: three storms with CHAMP orbits in the midday-midnight plane and three in the dawn-dusk plane (listed in Table 1). Five of the storms are described in the appendix / supplementary material to this paper.

The storm phases are identified in this study according to the SYM/H index, which describes together with the ASYM/H index the large-scale variations of the geomagnetic field with a 1-min

cadence. In essence, they represent mean values of the magnetic field deviation from the quiet time level for a longitudinally distributed chain of mid-latitude observatories. The current at the magnetopause (DCF) and the currents within the magnetosphere as the ring current (DR), which is symmetric with respect to the geomagnetic axis, and the tail current (DT), which closes via the dayside magnetopause, determine the intensity and the development of the magnetic disturbances.

These currents carry the main contributions to the SYM/H values during magnetic storm intervals (Maltsev, 2004; Alexeev et al., 1996).

The density of the ring current varies with longitude. This variability is identified as the partial ring current (PRC), denoted by the ASYM/H index, and determined as the difference between the maximum and minimum magnetic field values from a longitudinal chain of mid-latitude observato-

ries. The PRC current system is a 3-D one that is confined to a limited azimuthal range of the ring current in the magnetosphere, FACs between the magnetosphere and ionosphere at the border of the PRC, and an EE in the evening sector at ionospheric heights. The latitude position is controlled





by MLT and toward the near noon sector it shifts toward the ionospheric footpoint of the cusp region (Feldstein et al., 2006). The FAC of the PRC maps from ionospheric heights to the near-cusp

magnetopause region, where the Hall currents flow, which are controlled by the IMF By component.

## 4   The storm of 29–30 May 2003

The orbit of the CHAMP satellite in its ascending branch was on the dayside ($\sim$14–16 MLT), while its descending branch was in the nighttime sector ($\sim$02–04 MLT).

The beginning of the main magnetic storm phase was identified during orbit 16233 at 22:24 UT

(-61.6 nT), while the minimum value of SYM/H was recorded during orbit 16234 at 23:59 UT (-123.5 nT) and orbit 16235 at 01:33 UT (-139.5 nT). The four orbits prior to the main phase (16229–16232) at 16:18 UT–20:53 UT are characterized by SYM/H values of -1.6 nT, -35.6 nT, -59.6 nT, and -27.0 nT as well as the occurence of three substorms with intensities according to AL values in the range of $\sim$-1600 nT to $\sim$-2400 nT. ASYM/H increases sharply prior to the beginning of the

main phase (208 nT during orbit 16231) and during the beginning of the main phase (290 nT during orbit 16233). In the maximum of the main phase, the values of this index decrease to 75 nT during orbit 16234 and 145 nT during orbit 16235. Following the main phase, the recovery phase develops (orbits 16236–16240) at 03:03 UT–09:21 UT, in the course of which the SYM/H values return to the initial values at $\sim$60 nT and ASYM/H decreases to 51 nT during orbit 16239.

Let us now consider the structure and the latitudinal variation of the position and density of the electrojet during the various phases of the analysed storm. The eastward (EE) and westward electrojet (WE) can exist in the daytime sector at latitudes of the auroral zone($\sim$60°<MLat<70°), while poleward of it, at latitudes of the auroral oval ($\sim$73°<MLat<79°), the currents of the polar electrojet (PE) can appear. The direction of the PE, however, can be eastward or westward. This is determined

by the sign of the IMF By component: eastward current in the PE for By>0, and westward for By<0. In the nighttime sector, the current is westward directed (WE) in the majority of cases at auroral latitudes. Fig. 2 shows the direction, MLat, and density of the Hall currents along the orbit for dayside (left column) and nightside (right column) sectors as obtained from scalar measurements of the geomagnetic variations corresponding to the modelled current variations of Ritter et al. (2004b). It is

obvious that the quantity, locations, and intensity of the peaks along the latitudinal current density distribution vary in the course of the storm development.

### 4.1   Observations related to SYM/H variations

The latitudinal variation of the position and density of the EE is shown in Fig. 2. During the orbits 16229 and 16230, one singular peak of eastward current was observed, which occurred at

MLat = 63.4° and MLat = 64.0° with intensities of 1.0 $A/m$ and 0.6 $A/m$, respectively. This means that the EE peak current diminishes in density with increasing disturbances according to the SYM/H





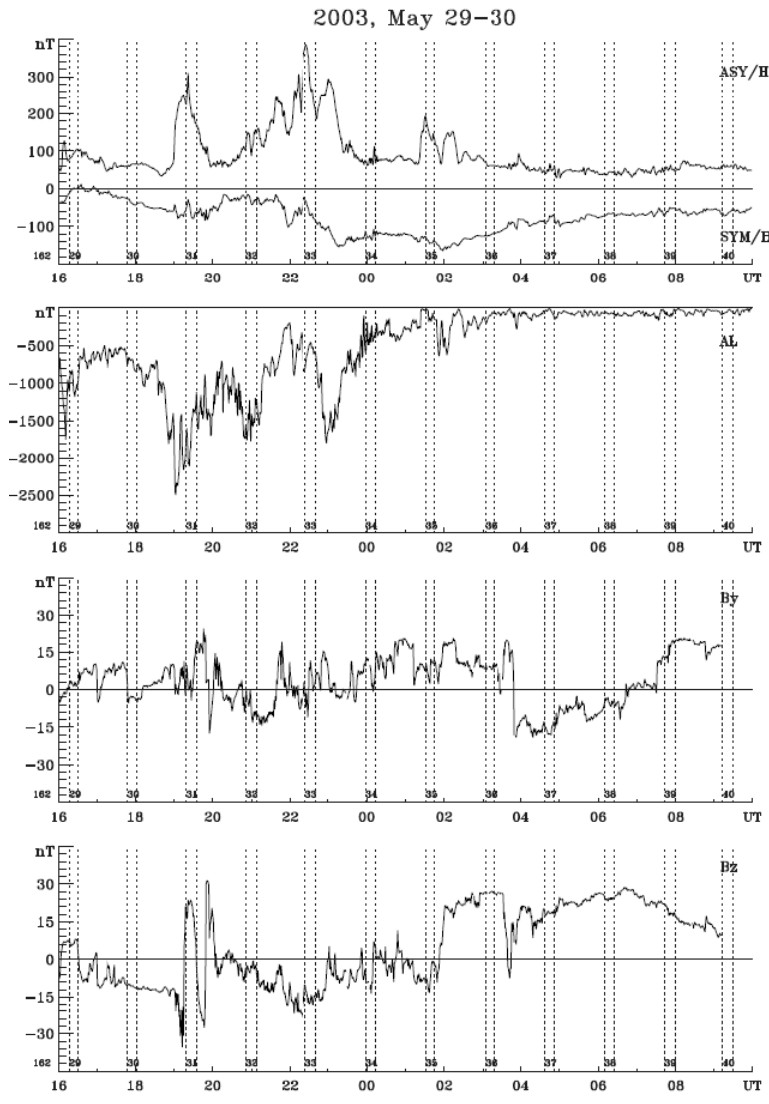

**Fig. 1.** One-minute values of the ASYM/H, SYM/H, and AL indices and of the $B_y$ and $B_z$ components of the IMF for the storm of 29–30 May 2003 (analysed interval from 16:00 UT on 29 May to 10:00 UT on 30 May 2003, orbits 16229–16240). The vertical dashed lines indicate the UT time moments of each satellite orbit over the northern polar cap. The orbit numbers are splitted into two parts: the two digits above the UT-axis of each frame denote the last two digits of the orbit numbers of the CHAMP passes, while the first three digits are indicated at the lower left side.

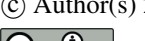


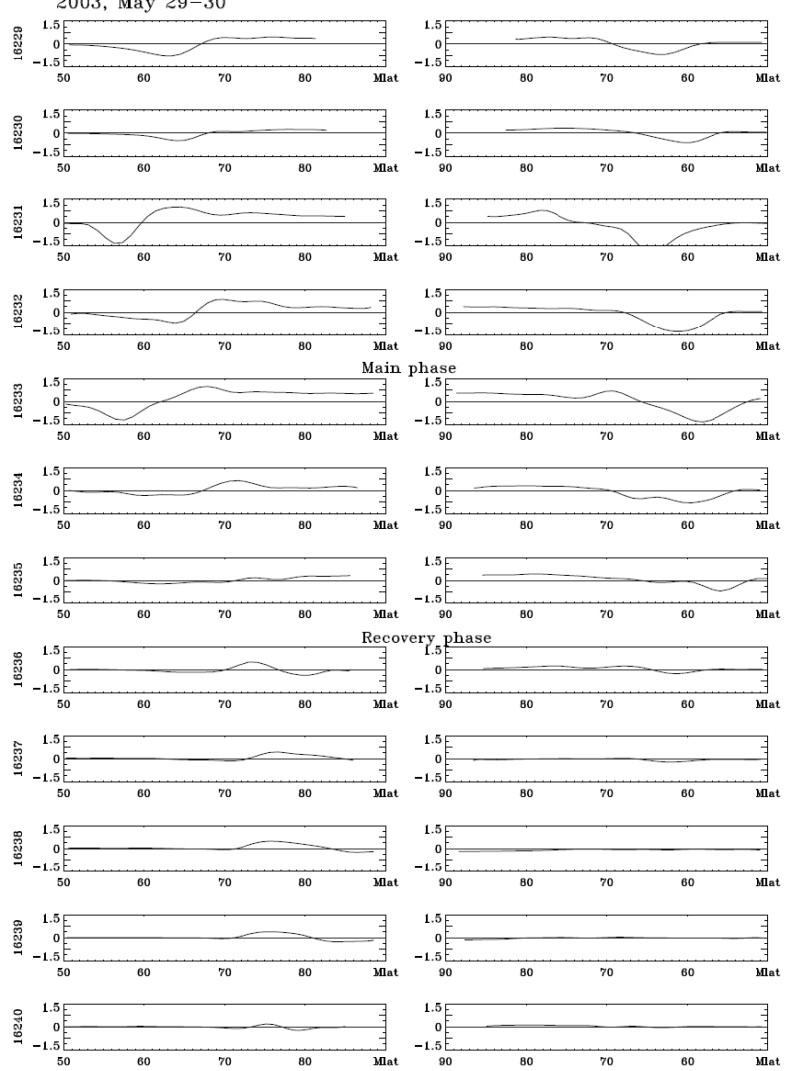

**Fig. 2.** Direction and density values of the Hall current along the satellite orbit at the dayside (left column, 14–16 MLT, corresponding to the ascending section of the orbit) and nightside sectors (right column, 02–04 MLT, descending orbit section). Positive currents denotes eastward current for the descending orbit section, and, accordingly, westward current for the ascending section.

index and shifts to higher latitudes. During the orbits 16231–16233 (in the substorms interval and at the beginning of the main storm phase), one can clearly note two intense peaks: one of the eastward current (EE), and another of the westward current (WE - the westward electrojet). The EE peak



during orbit 16231 amounts to ∼1.83 $A/m$ at MLat = 56.3°, decreasing in the course of the next orbit to 0.94 $A/m$ at MLat = 63.9°.

With the beginning of the main phase, the current again intensifies to 1.61 $A/m$ at MLat = 57.6°. The EE variations in density and latitudinal position during the orbits 16231–16233 proceed analogeous to the SYM/H changes: the more intense SYM/H, the closer to the equator shifts the EE peak

and the stronger becomes its current density.

But such an accordance is broken again in the maximum of the main phase similar to orbits 16229 and 16230: SYM/H increases in intensity, while the EE in its peak diminishes to 0.44 $A/m$ and even 0.29 $A/m$ at MLat = 61.0°. The EE peak amounts to 0.23 $A/m$ at MLat = 65.2° during orbit 16236 in the recovery phase and diminishes further to values <0.2 $A/m$ during the subsequent orbits,

which complicates the identification of the EE position. This way the EE follows with its varying current densities at auroral latitudes the creation and main phases of the magnetic storm. It should be noted, that SYM/H remains still significant during the recovery phase, with values of ∼-60 $nT$. This exceeds the intensity of SYM/H during the substorm interval and at the beginning of the storm main phase, while the EE during the recovery phase is much smaller than during the substorm interval.

In contrast to the EE, a westward current exists on the dayside sector during the whole interval considered, except the two first orbits, and achieves during the substorm interval values of 1.1–1.3 $A/m$ within the peaks at 64°<MLat<70°. It is obvious that the WE does not attain the CHAMP meridian (∼14–16 MLT) during the first two orbits, while it strengthens in the nighttime sector and propagates toward the evening hours. During the orbits 16231–16233, the latitude and density of

the WE peak changes in phase with the SYM/H intensity, analogous to the current variations in the EE peak. During the maximum of the main phase the WE peak diminishes to 0.8–0.25 $A/m$ at 71°<MLat<74° in antiphase to SYM/H.

The peaks of westward current remain at a level of ∼0.6 $A/m$ within 73°<MLat<76° during the recovery phase. An additional peak of eastward current with a density of 0.49 $A/m$ appears during

orbit 16236 at MLat = 80.5° during early afternoon hours (MLT∼14.5 hrs). This latitude and the MLT range around midday imply that the current observed is the PE. In this case its orientation is controlled by the IMF $B_y$ component and for an eastward PE the $B_y$ component should be positive (Friis-Christensen et al., 1972; Sumaruk and Feldstein, 1973; Feldstein, 1976). Indeed, according to Fig. 1, $B_y$= 9 $nT$ during the period of this orbit. During the two subsequent orbits, the Hall

current changes its direction to westward at MLat∼80°. If this westward current prove to be the PE, then its appearance should be connected with a change of the IMF By component. Indeed, the currents are accompanied with a change of sign of the IMF $B_y$ component, corresponding to -17.5 $nT$ (MLat = 80.7°) during orbit 16237 and -5.2 $nT$ (MLat =80.6°) during orbit 16238. During the orbits 16239 and 16240 the IMF $B_y$ component turns to positive values again and weak eastward

directed currents appear accordingly at MLat∼80°.

In the majority of latitudinal profiles of the nighttime sector (Figs. 2), there one peak of the WE





exists at latitudes of the auroral oval and some weakly spreaded eastward currents. Two orbits constitute an exception - one prior to (16231) and another during the beginning of the main storm phase (16233). These orbits pertain to the period of intense substorms. Within the polar cap up to

the geomagnetic pole, there exist quite intense (up to 0.9 $A/m$) eastward currents.

These currents might contain irregularities, which are caused by the appearance of a peak of eastward currents in the latitudinal profile. The monotonicity of the eastward current variations within the polar cap during most orbits provides some reason to assume, that these currents result from the closure of an intense WE current, which occurs at latitudes of the auroral oval in the

nighttime sector.

At the beginning of the substorm interval (orbits 16229 and 16230) with the intensification of SYM/H, the WE peak shifts to lower latitudes and the current density diminishes. The most intense peaks of the nighttime WE are obtained during the substorm interval prior to the main phase start with and retain values of 2.7 $A/m$ at MLat = 64.4° (orbit 16231), 1.7 $A/m$ at MLat = 60.9°

(orbit 16232), and during the beginning of the main phase with 1.79 $A/m$ at MLat = 58.0° (orbit 16233). Later in the maximum of the main storm phase, the WE peak current density diminishes to 1.07 $A/m$ at MLat = 59.5° (orbit 16234) and 0.9 $A/m$ at MLat = 55.8° (orbit 16235). Hence, the latitudinal peaks of the WE vary during nighttime in phase with the intensification of SYM/H (storm development) before the main phase commences at higher latitudes, while shifting to the equator

during the maximum of the main phase. The peak intensities changes both in phase and in antiphase with the SYM/H intensity. During the recovery phase, the peak density of the WE current is smaller than 0.2 $A/m$, while the eastward currents within the polar cap are too small to be recorded.

### 4.2   Observations related to ASYM/H variations and to high-latitude currents

In the dayside sector during the existence of the EE (orbits 16229–16236), the peak current intensi-

ties and the peak latitude positions vary synchronous with the ASYM/H changes, except of one orbit (16235) during the main phase. During this orbit, the ASYM/H index abruptly intensifies to 145 $nT$ with a pertaining small density of the EE with ∼0.29 $A/m$ and a shift of MLat by 1.8°. For the WE, the change in latitude and density of the peak currents is in phase with the ASYM/H variations during the storm, with the exception of orbit 16235.

In the nighttime sector, the intensity of the peaks and their latitude (except orbit 16235) change in phase with the ASYM/H variations.

Summarizing the results of Hall current observations by the CHAMP satellite during the magnetic disturbance period of 29–30 May 2003 in the daytime and nighttime sectors (12–16 MLT and 00–04 MLT, respectively) we conclude:

– Intense >1 A/m eastward and westward electrojets can occur at latitudes of the auroral zone during substorm periods, which precede the magnetic storm, and during the beginning of its main phase. During the maximum of the main phase, the density of the Hall currents as well





as the substorms diminish in antiphase to an increase of the SYM/H index.

- A fast decay of the EE and WE occurs during the recovery phase at auroral latitudes both during daytime and nighttime hours. The westward or the eastward currents can be influenced during this storm phase by the existence of a PE at $73°<$MLat$<80°$ in the region of the dayside cusp.

- The direction of the current in the PE is determined by the IMF $B_y$ component: for $B_y > 0$ the current is eastward, for $B_y < 0$ westward. The change of the current direction within the PE can occur several times during the storm development, but always in accordance with the change of the IMF $B_y$ orientation.

- The Hall currents in the auroral ionosphere, both the EE and the WE, vary usually in phase with the SYM/H and ASYM/H variations (but sometimes also in antiphase). There are time intervals, where any correlation between the geomagnetic activity indices and the Hall current parameters is missing. There is a closer connection of the current density and the MLat variations with ASYM/H than with SYM/H.

- In the daytime sector (14–16 MLT) during a period of intense substorms, the EE is located in a latitude range $56°<$MLat$<64°$, while the WE is at $64°<$MLat$<70°$. During the main phase of the storm, the EE shifts to $58°<$MLat$<62°$, while the WE is situated at $64°<$MLat$<73°$, and during the recovery phase, finally, the WE is observed at latitudes of $73°<$MLat$<76°$. Therefore, the EE stays at about the same latitudes during the both the intense substorms and the main phase of the storm, attaining extreme equatorward values of MLat$\sim 56°$. An analogue situation exists with regard to the change of position for the WE in various storm phases, but during daytime hours the WE is located about $6°$ closer to the pole.

- In the nighttime sector (02–04 MLT), there exists practically only the WE, which is located during substorms at $61°<$MLat$<64°$, and during the main storm phase at $56°<$MLat$<60°$. Therefore, extremal positions of the WE and EE can reach latitudes below $60°$. This occurs in the daytime sector for the EE, while in the nighttime for the WE.

The detailed description of the Hall current dynamics during five further magnetic summer storm intervals is transferred to the Appendix.

## 5 Discussion

In the section 4 and in the appendices A1–A5 we have investigated several geomagnetic storm periods, based on magnetometer measurements onboard the CHAMP satellite. The Hall currents in the high-latitude upper ionosphere of the Northern Hemisphere were analysed for various MLT sectors



with regard to their position in geomagnetic latitude, their density and direction. The empirical de-
scription concerned the appearance of the EE, the WE, and the PE during various storm phases and
was carried out primarily qualitatively.

Below we are going to analyse the current directions, their densities, and MLat positions for
various MLT sectors with regard to solar wind parameters and some indices of the planetary magnetic
activity (SYM/H, ASYM/H, AL, IndN). We use activity indices, which characterize the occurrence
and dynamics of large-scale plasma domains in Earth's magnetosphere that are responsible for the
existence of concrete variations of the geomagnetic field at Earth's surface.

### 5.1 Polar electrojets

It is well-known from geomagnetic activity researches that the intense magnetic disturbances at the
high-latitude projection of the magnetospheric cusp are not related to the occurrence and dynamics
of magnetospheric substorms. Wang et al. (2008) investigated the behaviour of electrojets by means
of CHAMP satellite observations during two magnetic storm events with the focus on the auroral
electrojets.

Fig. 3a–f shows the magnetic latitude (left side panels) and the Hall current density $I$ (right side
panels) obtained by CHAMP satellite crossings over the polar electrojets during 6 geomagnetic
storms. The direction of the Hall currents can be distinguished in the upper panels (Fig. 3a): west-
ward and eastward currents are indicated with blue and red data points, respectively. For the further
study, we selected the electrojet parameters at their extremal values of current density for each or-
bit. The data pool was augmented yet be including also neighbouring values before and after the
extremal points.

**Table 2.** The dependent ($X$) and independent variable ($Y$), their correlation coefficients ($r$), the coefficients $A$
and $B$ of the regression equations $X = A + B * Y$, and their dispersions $\sigma$.

| X | Y | $r$ | A | B | $\sigma$ |
|---|---|---|---|---|---|
| I (density, A/m) | By (>0) | 0.59 | 0.535 | 0.018 | 0.160 |
| I (density, A/m) | By (<0) | -0.72 | 0.291 | -0.024 | 0.134 |
| I (density, A/m) | $|By|$ | 0.56 | 0.433 | 0.018 | 0.170 |
| I (density, A/m) | ASYM/H | 0.74 | 0.396 | 0.004 | 0.138 |
| MLat (deg) | AL | 0.46 | 78.540 | 0.006 | 2.542 |
| MLat (deg) | IndN | -0.52 | 77.750 | -0.006 | 2.415 |

Fig. 3a differentiates the current measurements with regard to the azimuthal IMF component (By),
i.e., between those, obtained during By>0 and those during By<0 conditions. It is clearly seen that
the direction of the current within the PE is determined by the IMF By sign. For intervals with
positive IMF By>0, we observe for all cases an eastward directed Hall current; for negative IMF





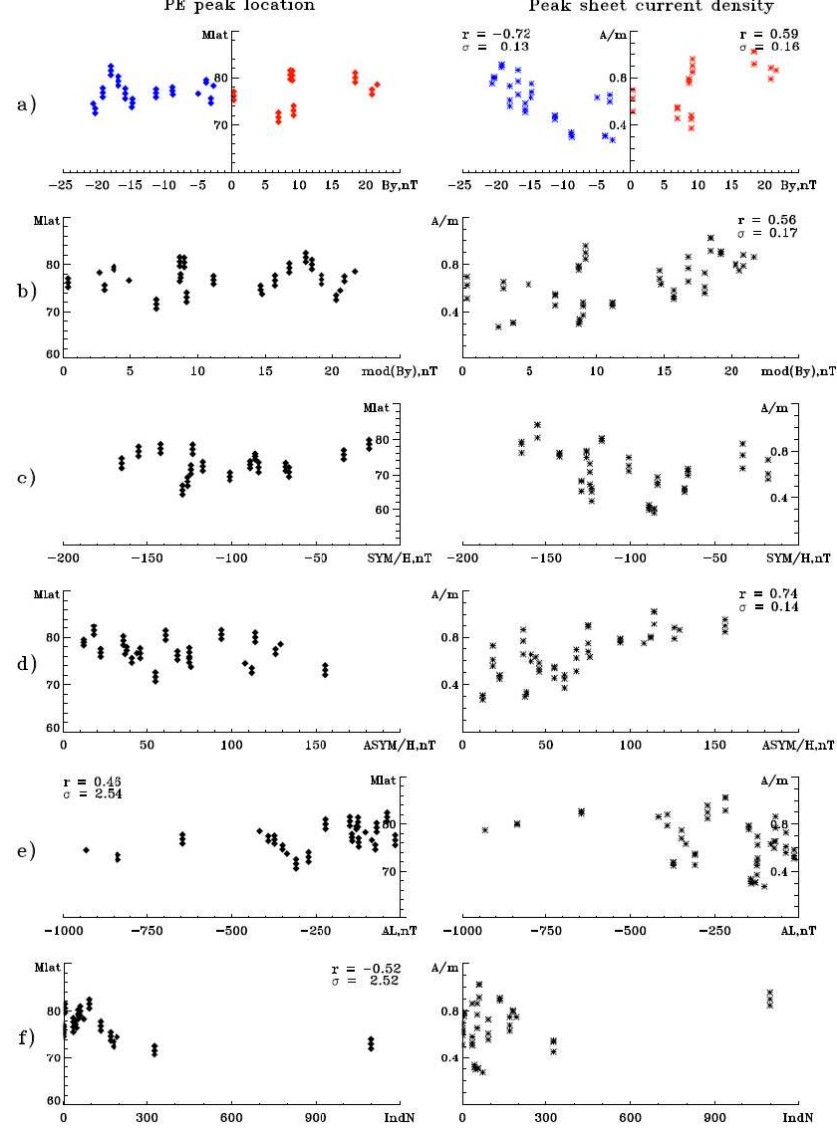

**Fig. 3.** Dependence of the magnetic latitude MLat (degrees) position of the peak (left column) and its density ($I$ in A/m, right column) of the Hall current in the polar electrojet (PE) on the IMF By component (a) and its magnitude (b), on the geomagnetic activity indices SYM/H (c), ASYM/H (d), AL (e), and the solar wind coupling function IndN (f). For the cases of correlations with $r > 0.46$, the correlation coefficients ($r$) and the dispersion ($\sigma$) according to a linear regression are shown as labels.





By<0 intervals the Hall current is always westward. The current density within the PE is correlated
with the magnitude of the azimuthal IMF component By: it increases from 0.3–0.4 A/m for near-
zero values to $I \sim$0.9 A/m for $|B_y| \sim$23 nT (i.e., the maximum By value during the period shown).
Correlation coefficients $r$ between current density $I$ and the IMF By component for By>0 and
By<0 are shown in the upper right and upper left corner, respectively (Fig. 3a, right panel). The

coefficients for the offset ($A$) and the slope ($B$) of the linear regression line (for $r \geq 0.46$) as well as
the dispersion values $\sigma$ are listed in Table 2.

The current density values $I$ for $B_y \sim 0$ nT are somehow different in the regression equations
for $B_y > 0$ and $B_y < 0$. Fig. 3b (right panel) shows the current density $I$ as a function of the
IMF component's magnitude $|B_y|$, i.e., independent of the IMF By sign. This increases the number

of data points for linear regression estimation. According to this estimation, the current density
amounts to $I \sim$0.4 A/m for $|B_y| \approx$0 nT, while it attains $\sim$0.9 A/m for $|B_y|$ =23 nT, i.e., about twice
as large. The increase of the Hall current density within the PE might take place during magnetically
quiet intervals during the absence of magnetic activity at latitudes of the auroral zone, which is not
directly related to the IMF By component.

We could not find any essential correlation between the MLat position and the IMF By compo-
nent for both $B_y > 0$ ($r = 0.04$) and $B_y < 0$ ($r = 0.32$) nor for $|B_y|$ ($r = 0.15$, see Fig. 3a and
b). The current density and its direction (eastward or westward) within the PE is controlled by the
IMF By component, but the latitudinal position of the current density maximum does not depend on
IMF By. The observed morphological peculiarity of the PE is caused by its generation mechanism.

This is assumed to be due to the interaction between the magnetosphere and the supersonic plasma
flow (solar wind) with a "frozen-in" magnetic field (IMF). The PE currents are generated at mag-
netic latitudes of the cusp due to reconnection processes between the IMF and the geomagnetic field
(Jørgensen et al., 1972; Wilhjelm and Friis-Christensen, 1971). The reconnection of magnetic fields
brings about a north-south electric field and an east-west Hall current at cusp latitudes in the iono-

sphere. A possible generation mechanism for the PE current system has been suggested by Leontyev
and Lyatsky (1974), including the structure of the PE current system.

Leontyev and Lyatsky (1974) postulate the penetration of the electric field $E_z = V_x \times B_y$, where
$V_x$ is the solar wind velocity past the magnetosphere. This electric field will cause a potential
difference $U$ between the northern and southern boundaries of the magnetotail: $U = E_z \times D_m$,

where $D_m$ is the size of the magnetosphere along the z-axis. On the assumption of high conductivity
along the magnetic field lines, the electric field $E_z$ can exist only in the region of open field lines,
rooted at the polar caps, and will be short-circuited along closed field lines. Thus, the boundary
between closed and open field lines (OCB) will be the line of zero potential for the field $E_z$, and
$U$ =0 at this boundary in the ionosphere.

Feldstein et al. (1975b) estimated the effectiveness of the electric field penetration from the solar
wind to the cusp. During summer season (July–August 1969) the integral Hall current within the PE



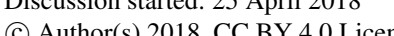


was estimated to $\sim$220 kA. The integral conductivity in the ionosphere during summer around noon is $\sim$7 mhos, i.e., the potential drop in the cusp is about $U \sim$30 kV. The potential difference between the northern and southern boundaries of the magnetotail amount to $U_m \sim$600 kV. The following assumptions were made for the estimation: solar wind speed $V_x \sim$400 km/s, IMF $B_y =$6 nT, and the magnetospheric size along the z-axis $\sim$40 $R_E$. If the voltage drop is the same in the northern and southern hemispheres, then the efficiency of the electric field penetration from the solar wind to the high-latitude ionosphere during summer season is $\sim$10%.

Here, we present estimations of the efficiency of the electric field penetration into the cusp region for two concrete orbits of the CHAMP satellite over the daytime sector.

During orbit 16238 of 30 May 2003 we observe a maximum Hall current density at 06:15 UT. The solar wind velocity at this time was $V_x =$640.7 km/s, the plasma density in the solar wind $n_p =$21.4 cm$^{-3}$, and IMF $B_y =$-6 nT. The electric field in the solar wind amounts to $E_z = V_x \times B_y =$3.8 mV/m. The dynamic plasma pressure at the subsolar point $P_{sw} = 0.88P_{sw}^{dyn} =$12.9 nPa, while the distance of the subsolar point at the magnetopause is $\sim$7 $R_E$. The potential difference along the z-axis between the northern and southern tail boundaries amounts to $U_m =$338 kV, and in the cusp of one hemisphere hence $U_m =$169 kV. According to the CHAMP data, the mean density of the current is $\sim$0.3 A/m over $\sim$8.5°, the integral current in the cusp therefore $\sim$295 kA, and the potential difference in the cusp $\sim$42 kV. The efficiency of the electric field penetration from the solar wind into the ionosphere is thus about $\sim$25%.

During orbit 29018 of 24 Aug 2005, 16:42 UT, with $V_x =$630.5 km/s, $n_p =$18.6 cm$^{-3}$, and IMF $B_y =$-20.9 nT results in $E_z =$12.6 mV/m, $P_{sw} =$10.8 nPa, and distance of the subsolar point at the magnetopause is $\sim$7.3 $R_E$. The potential difference in the cusp of one hemisphere can thus be estimated to $U_m =$585 kV. According to the CHAMP data, the mean density of the current is $\sim$0.4 A/m over $\sim$7.5°, the integral current in the cusp therefore $\sim$348 kA, and the potential difference in the cusp $\sim$49 kV. The efficiency of the electric field penetration from the solar wind into the ionosphere is thus about $\sim$9%. Therefore, for a quite variable electric field voltage applied to the magnetosphere from the magnetized solar wind flow (from 585 kV to 169 kV), the efficiency of its penetration to the ionosphere varies between 9% and 25%.

The SYM/H index varied during the intervals of CHAMP overflights above the polar electrojets considered in this study from -10 nT to -170 nT. As shown in Fig. 3c, there is no correlation between SYM/H and the MLat positions ($r = -0.13$) nor the PE current density ($r = -0.01$). The absence of any correlation is as expected, because the current systems of the DCF, DR, and DT are located completely within the magnetosphere. In case of absent FACs, they cannot serve as sources for Hall currents in the ionosphere that is responsible for the existence of PE.

According to Fig. 3d, there is a high correlation between the ASYM/H index and the PE current $I$ ($r \sim 0.74$) and an absence of correlation with the MLat position of the PE ($r \sim -0.29$). The PE current density has therefore a direct relation to the intensity of the ASYM/H current system: with



increasing longitudinal asymmetry increases the PE current density, but the latitudinal position of the PE does not depend on ASYM/H.

Fig. 3e shows the correlation between the MLAT position and the density of the PE with the AL index of geomagnetic activity. This index describes the reduction of the horizontal component of the geomagnetic field at auroral latitudes on Earth's surface during disturbances with respect to quiet-time conditions, using a longitudinal chain of magnetic observatories. The AL index appears to be a sensitive tracer for processes in the central plasma sheet of the magnetospheric tail. These processes are created by injection of energetic particles, their accumulation, and the dissipation of their energy during storm times and is accompanied by changes of the boundary positions of large-scale plasma structures. They appear to have relatively small influence on the density and MLat position of the PE (with $r = -0.38$ and $r = 0.46$, respectively). However, there is a distinctive tendency for the shift of the PE from $\sim78°$ to $\sim74°$ with an increase of the AL index up to -900 nT.

As shown in Fig. 3f, there is a correlation of IndN with MLat in the daytime sector (r=-0.52). This is obvious, because both components By and Bz are included in the definition of IndN. With increasing IndN, the latitude of the current decreases. The correlation coefficient of IndN with the Hall current density ($I$) is r=0.3.

## 5.2 Auroral electrojets

The most intense Hall currents at ionospheric heights, which are responsible for the electrojets, are located at auroral latitudes in the nighttime hours. It is even there, where intense auroras occur most often in the zenith (Chapman and Bartels, 1940; Harang, 1951). These electrojets were named auroral electrojets (AE). A huge number of studies has been published on their morphology, their connections with the solar wind parameters and the plasma domains in Earth's magnetosphere, as well as on their internal processes. The AE are present during all hours of the day. The number of electrojets, their internal current structure, and the interconnection with the individual magneto-spheric plasma domains depends both on the activity level and on the MLT position of the observation (Feldstein et al., 2006). Therefore, we consider below the results of the Hall current observations of the CHAMP satellite separately for each of the following four MLT sectors: daytime, nighttime, evening, and morning hours.

Figures 4–7 consider the MLat positions (left columns) and current densitites $I$ (right columns) during the moments of extreme values of current density in dependence on the SYM/H, ASYM/H, AL, and IndN indices. As in Fig. 3a, data points of electrojets with an eastward direction are indicated by red colour and those with westward direction by blue colour.

Table 3 provides the correlation coefficients $r$, the coefficients $A$ and $B$ of the linear regression equations of the type $X = A + B * Y$, which were obtained by the least-squares method with correlation coefficients $r > 0.46$, and the mean-square deviation $\sigma$ from the regression line.



**Table 3.** The dependent ($X$) and the independent variable ($Y$), their correlation coefficients ($r$), the coefficients $A$ and $B$ of the regression equations $X = A + B * Y$, and their dispersions $\sigma$, listed for four different MLT intervals.

| X | Y | $r$ | A | B | $\sigma$ |
|---|---|---|---|---|---|
| **MLT 09:00–13:59** | | | | | |
| MLat (WE, deg) | ASYM/H | -0.54 | 74.136 | -0.052 | 3.89 |
| MLat (EE, deg) | ASYM/H | -0.49 | 70.327 | -0.047 | 3.91 |
| MLat (EE, deg) | AL | 0.68 | 70.192 | 0.005 | 3.28 |
| MLat (WE, deg) | IndN | -0.74 | 72.215 | -0.011 | 3.13 |
| MLat (EE, deg) | IndN | -0.67 | 70.271 | -0.025 | 3.43 |
| **MLT 14:00–20:59** | | | | | |
| MLat (WE, deg) | SYM/H | 0.49 | 72.783 | 0.041 | 4.23 |
| MLat (EE, deg) | ASYM/H | -0.54 | 65.971 | -0.036 | 3.67 |
| Intensity (WE, A/m) | ASYM/H | 0.68 | 0.168 | 0.003 | 0.24 |
| Intensity (EE, A/m) | ASYM/H | 0.64 | 0.217 | 0.004 | 0.29 |
| MLat (EE, deg) | AL | 0.46 | 64.707 | 0.004 | 3.88 |
| Intensity (EE, A/m) | AL | -0.59 | 0.320 | -0.001 | 0.31 |
| **MLT 21:00–01:59** | | | | | |
| MLat (WE, deg) | SYM/H | 0.53 | 63.806 | 0.032 | 2.25 |
| Intensity (WE, A/m) | ASYM/H | 0.50 | 0.205 | 0.005 | 0.33 |
| Intensity (WE, A/m) | AL | -0.67 | 0.233 | -0.001 | 0.28 |
| Intensity (WE, A/m) | IndN | 0.76 | 0.207 | 0.003 | 0.25 |
| **MLT 02:00–08:59** | | | | | |
| MLat (WE, deg) | SYM/H | 0.47 | 67.343 | 0.040 | 3.24 |
| Intensity (WE, A/m) | ASYM/H | 0.69 | -0.089 | 0.010 | 0.38 |
| Intensity (WE, A/m) | AL | -0.52 | 0.328 | -0.001 | 0.44 |

### 5.2.1 Daytime sector 09:00–14:00 MLT

The AE in the daytime sector can coexist with the PE. These two types of current can be distinguished according to the following indications (that are valid for AE in contrast to PE):

1. The AE are as a rule located at MLat<73° during low geomagnetic activity conditions;

2. the Hall current direction in the AE does not depend uniquely from the orientation of the IMF By component.

Fig. 4 shows only those cases of AE appearance in the daytime sector with changing SYM/H



index. Usually, SYM/H has negative values (SYM/H<0 nT) during geomagnetic storms. Fig. 4a shows, however, beside of the mostly negative values also some values with SYM/H>0. They occur as a rule during the first few hours of magnetic storms. The large scatter of the data points and their low correlation coefficients (maximum for MLat(EE, $r = 0.39$) and $I$(EE, $r = 0.29$) in Fig. 4a)

indicate the weak control of the AE parameters by the symmetric ring current, the index of which is SYM/H.

The MLat position of the AE in the daytime sector correlates with three other indices: ASYM/H, AL, and IndN. The AE shifts with increasing disturbances toward lower latitudes: the WE from 72° to 66°, and the EE from 70° to 57° (Fig. 4b–d). The largest correlations of MLat are found with the

IndN coupling function (WE, $r = -0.74$), the smallest values for ASYM/H (EE, $r = -0.49$). These three indices characterize the large-scale current systems, the magnetic fields of which influence the magnetic field configuration of the dayside sector. It should be noted that there are tendencies for the WE to be located during daytime hours a few degrees more poleward than the EE. These tendencies are clearly visible with regard to the MLat(EE and WE) positions and their relation to ASYM/H and IndN (Fig. 4b and d). The constant term $A$ is in the case of ASYM/H 3.8° larger for the WE

than for the EE, and 1.9° in the case of the solar wind coupling function IndN. The correlation coefficients for the Hall current with the IMF By vector component and its magnitude is low (not shown). A significant correlation coefficient $|r| > 0.49$ is achieved in the daytime sector only for the MLat positions of the electrojets, while the correlation with the current densities is minimal. The electrojets can be both westward and eastward. The EE can be observed for very intense disturbances

during the storm period down to MLat∼57°.

### 5.2.2  Evening sector 14:00–21:00 MLT

Significant correlation values $r$ exist in the evening sector for both the current densities and the MLat positions of the electrojets. The largest values of $r \sim 0.6 - 0.7$ were obtained for current

densities $I$, independent of the current directions (westward or eastward) in the electrojets. There appears a dependence of MLat(WE) from the SYM/H index: the electorjets shifts equatorward with an increase of the ring current. The EE is located more equatorward than the WE by about ∼6°. The constant term $A$ of the regression equations amounts accordingly to MLat(EE)∼66° with respect to ASYM/H and MLat(WE)∼72° with respect to SYM/H. The EE current density exceeds those of the

WE. That means, the interpretation of the EE in the evening sector as a branch-off from the WE at higher latitudes will become more unlikely. The electrojets move more equatorward with increasing disturbance level according to any geomagnetic activity index. Their current densities rise from <0.2 A/m to 1.6 A/m for the EE, and up to 1.3 A/m for the WE. The EE is observed equatorward of MLat∼60° during magnetic storm periods with the threshold latitude for the EE shift of ∼53°.





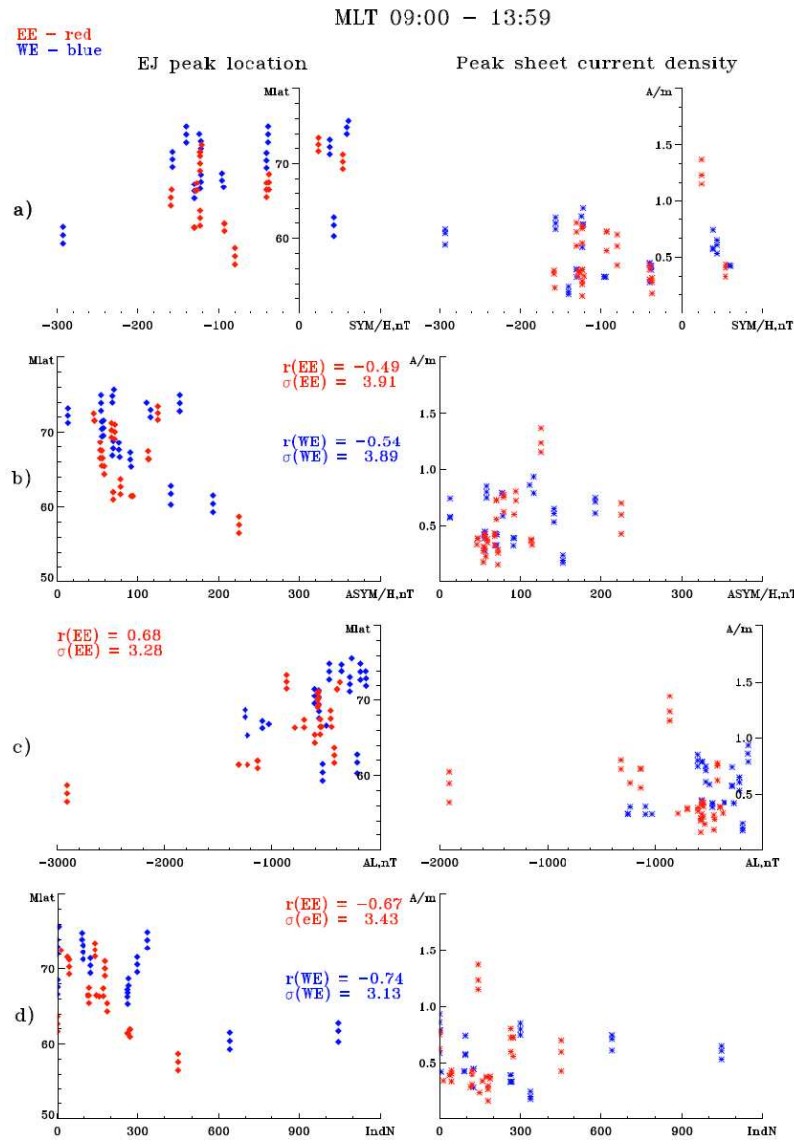

**Fig. 4.** Daytime sector (09–14 MLT): Dependence of the magnetic latitude MLat (degrees) position of the peak(left column) and of the density ($I$ in A/m, right column) of Hall current in the WE (blue) and EE (red) on the geomagnetic activity indices SYM/H (a), ASYM/H (b), AL (c), and the solar wind coupling function IndN (d). For the cases of correlations with $r > 0.46$, the correlation coefficients ($r$) and the dispersion ($\sigma$) according to a linear regression are shown as labels.





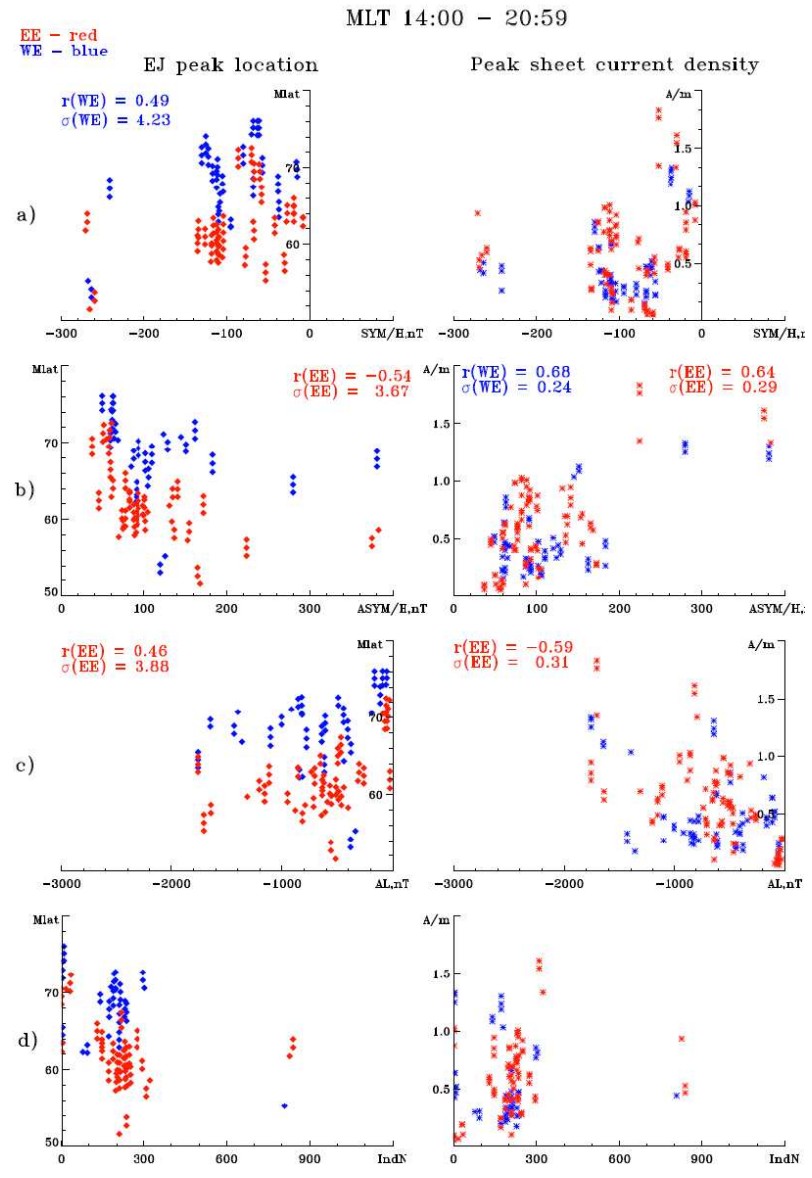

**Fig. 5.** The same as in Fig. 4, but for the evening sector (14–21 MLT).

5.2.3   Midnight sector 21:00–02:00 MLT

In this sector, the WE exists almost exclusively (Fig. 6). Moreover, the current density correlates
here well with the ASYM/H, AL, and IndN indices with a maximum value of $r = 0.76$ for the IndN





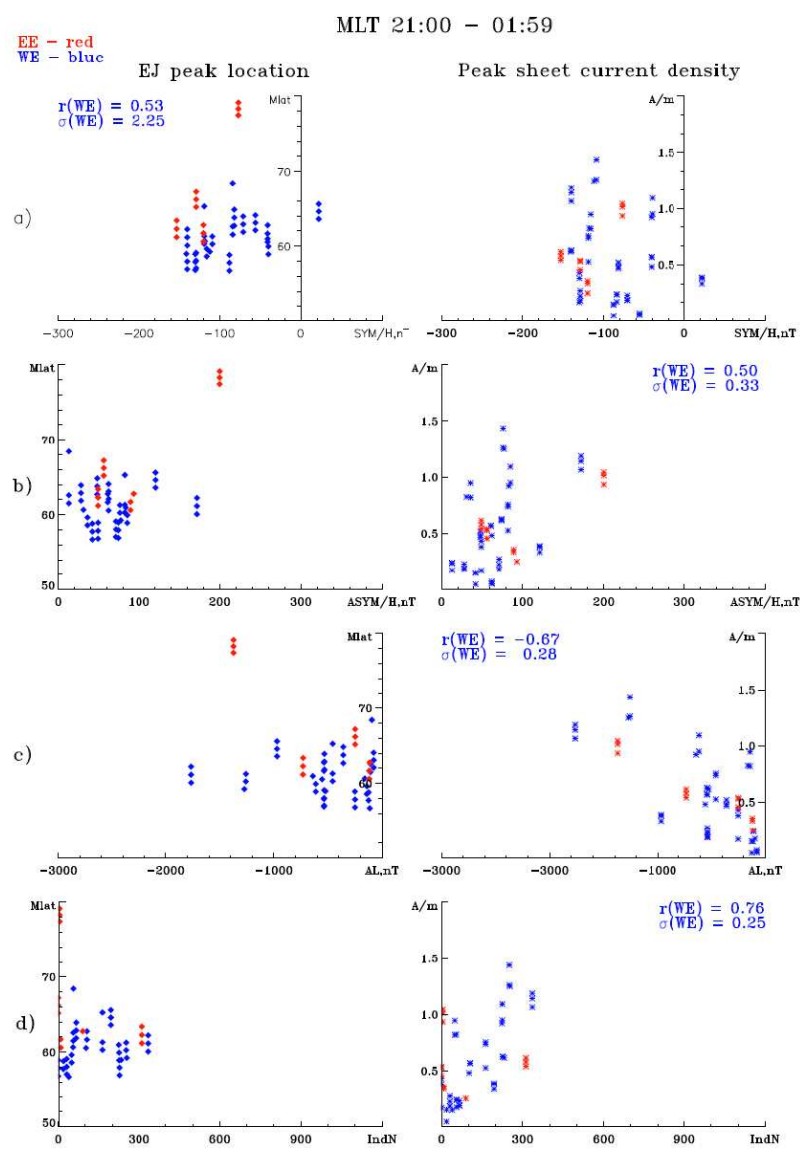

**Fig. 6.** The same as in Fig. 4, but for the midnight sector (21–02 MLT).

coupling function.

Otherwise, the MLat(WE) position correlates only with the SYM/H index. It decreases from
62° to 58° for a change of SYM/H from ∼-40 nT to ∼-170 nT. The WE current density increases
from values <0.2 A/m to ∼1.5 A/m for an intensification of the disturbance according to the IndN




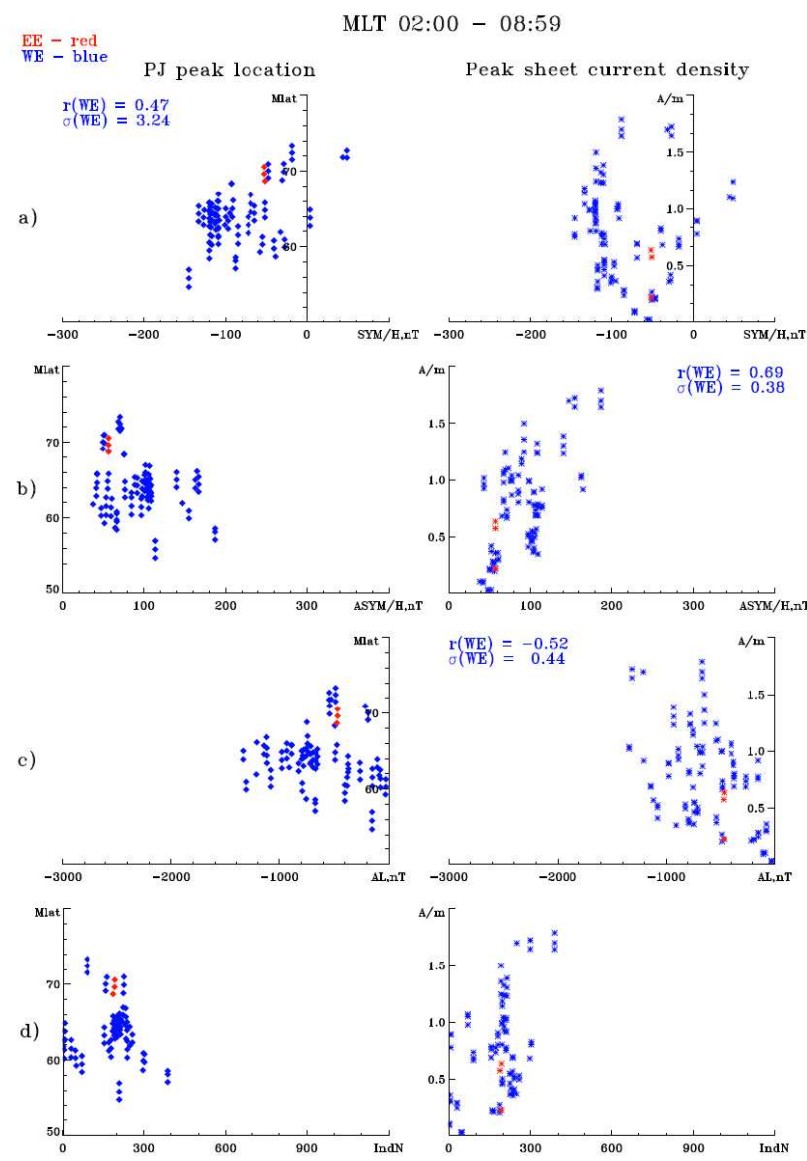

**Fig. 7.** The same as in Fig. 4, but for the morning sector (02–09 MLT).

coupling function from 0 to 325, while the WE position moves equatorward until a threshold value of ~58°.



### 5.2.4 Morning sector 02:00–09:00 MLT

Similar to the midnight sector, the WE exists also almost exclusively within the morning sector
(Fig. 7). The current density correlates here well with the ASYM/H and AL, with a maximum value
of $r = 0.69$ for the ASYM/H index. The WE current density increases from values 0.32 A/m to
1.92 A/m with an increase of the ASYM/H value from 40 nT to 200 nT, while the electrojet position
moves equatorward until a threshold value of $\sim 56°$. The MLat(WE) position correlates only with

the SYM/H index. In this regard the midnight and morning sectors show the same behaviour. The
MLat positions are controlled predominantly by the SYM/H index, i.e., by the density of the ring
current DR rather than by any other current system. The central plasma sheet of the magnetospheric
tail is the source region of the WE in the nighttime sector. An increase of the DR is accompanied by
a change of the geometry of the magnetic field lines that are interconnected with the central plasma

sheet. This results in a shift of the ionospheric projection of the WE toward the equator. The relation
between the current density in the WE and the AL value is not needed for the interpretation.

## 6 Conclusions

In this paper we investigated the density and spatial-temporal distribution (versus magnetic latitude
MLat and MLT) of Hall currents at high latitudes. The currents were determined from measurements

of total magnetic field data, sampled by magnetometers on board the CHAMP satellite at ionospheric
altitudes of $\sim 430$ km (Ritter et al., 2004a). In this study we used these current estimations to explore
the dynamics of the polar and auroral electrojets during a selection of six magnetic storms (see
Table 1). We identified their distinctive features and the correlations with activity indices that are
usually used to characterize large-scale current systems in the magnetosphere. The main findings

obtained are listed below.

### 6.1 Variations of the Hall currents during storms

- The characteristics and density of Hall currents change in the course of geomagnetic storms.
  Their structure correspond basically to the well-known characteristics and dynamics of elec-
  trojets in MLat and MLT during magnetic storms. A splitting of the WE is possible in the

morning hours during the recovery phase, analogous to the splitting of auroral luminescence
  in the auroral oval. These are additional, though indirect affirmations for the applicability to
  use magnetic field measurements at altitudes above the main ionospheric current layer for the
  determination of currents in the upper ionosphere.

- Substorms occuring prior to or during the beginning of the main phase of a storm are accom-

panied by an EE at auroral latitudes (MLat $\sim 64°$) during daytime MLT hours. Later they
  appear as WE both in the afternoon ($64° <$ MLat $< 70°$) and during nighttime (MLat $\sim 64°$).





- With the development of the main phase both the daytime EE and the nighttime WE shift to subauroral latitudes MLat~56°, while increasing in density up to $I$ ~1.5 A/m. Both electrojets exist during daytime and evening hours in the main storm phase. During evening hours, the WE is located by ~6°closer to the pole than the EE, and about 2°-3° during daytime hours.

- The current densities of EE and WE decrease quickly during the recovery phase, i.e., the electrojets vanish, but in the daytime sector at MLat~73°-80° appears a PE (polar electrojet) with a westward or an eastward direction, depending on the orientation of the IMF By component. The PE is eastward directed for By>0 and westward directed for By<0. Changes of current flow direction in the PE can occur manifold during the storm, but only due to changes of the IMF By orientation.

## 6.2   Hall current in the polar electrojet

While auroral electrojets are present at all local time hours, the PE is confined to daytime hours. Fig. 3a–f shows the results of the correlation analysis of the PE characteristics and the IMF By component as well as various activity indices. The values of the correlation coefficients $r$ and the coefficients $A$ and $B$ of the regression equations are listed in Table 2. They relate the current density and their MLat position to the indices that characterize the situation in the solar wind and within the magnetosphere at the time of the observations.

The PE currents and their MLat positions are characterized by the following peculiarities:

- The PE appears at magnetic latitudes and local times of the cusp.

- The direction of the current in the PE is controlled by the IMF By (azimuthal) component: for By>0 the current is eastward, for By<0 the current is westward directed.

- The current density in the PE increases with the intensity of the IMF By component from I~0.4 A/m for By~0 nT up to I~1.0 A/m for By~23 nT.

- The MLat position of the PE does not depend on the orientation and the strength of the IMF By component.

- Assuming that the penetration of the solar wind electric field into the cusp causes the generation of the PE, we estimate the efficiency of such a penetration. Based on two CHAMP orbits across the dayside sector of the high-latitude ionosphere, we estimate the potential difference over the cusp with 169 kV and 585 kV. The efficiency of the electric field penetration into the cusp would then amount to 25% and 9%, respectively.

- There is no connection between MLat and the current density $I$ in the PE with the magnetospheric ring current DR (index SYM/H).



- There is a correlation between the current density $I$ in the PE and the density of the partial
ring current in the magnetosphere (PRC, index ASYM/H), but practically no correlation of
this index with MLat of the PE.

- The currents in the central plasma sheet appear to have a weak influence on the current density
and the MLat position of the cusp.

- We realized a correlation between MLat and the IndN solar wind coupling function.

6.3   Hall current in auroral electrojets

Auroral electrojets are located at auroral latitudes (MLat<72° during daytime hours, and MLat<68°
during nighttime) exist during all MLT. The amount of electrojet current in a certain latitude range,
the structure of the currents in them, the interconnection with concrete magnetospheric domains,
depends on the level of disturbance, which is controlled by UT as well as local time (MLT) at the
observational points. Therefore we present the conclusions from the observations for each of the
four time sectors: daytime, evening, nighttime, and morning hours.

Daytime sector (09–14 MLT):

- The MLat positions of the auroral electrojets, both WE and EE, correlate with the activity
indices ASYM/H, AL, and IndN. The auroral electrojets shift toward lower latitudes with
increasing activity. For ASYM/H ∼220 nT the EE shifts to MLAT ∼57°.

- MLat(EE) collocates 3.8° equatorward of MLat(WE) according to the ASYM/H index and
1.9° according to the IndN coupling function.

- Significant correlation coefficients with $r > 0.49$ are obtained only for MLat, correlations
with the current density $I$ are, however, very small for all indices.

Evening sector (14–21 MLT):

- Significant values of the correlation coefficients $r$ with activity indices exist both for MLat
and for the Hall current density $I$.

- The largest correlation coefficients ($r$ ∼0.6–0.7) exist between the ASYM/H index and the
current density $I$(WE, EE).

- The EE shifts ∼6° more equatorward compared to the WE.

- The EE and WE shift equatorward with increasing activity. This shift occurs with respect to all
activity indices inspected here. The EE is located equatorward of MLat ∼60° during magnetic
storm periods; the farthest shift attains MLat ∼53°.



– The equatorward shift for increasing activity is accompanied by increasing current densities
from $I < 0.2$A/m for ASYM/H$\sim$40 nT to $\sim$1.7 A/m in the EE and $\sim$1.3 A/m in the WE for
ASYM/H$\sim$380 nT. The current density of the EE increases hence stronger than that of the WE
(by about 30%).

– The current density in the EE is larger than in the WE. The WE is missing for a certain
confined MLT interval of the evening sector, in case of an existing EE (Feldstein et al., 2006).
That means that the EE in the evening sector cannot be a low-latitude branch-off from the WE
current.

Near-midnight sector (21–02 MLT):

– Around midnight, the WE is predominant.

– The current density in the WE correlates with the activity indices ASYM/H, AL, and IndN,
with a maximum correlation coefficient of $r \sim$0.76 for the IndN.

– The MLat(WE) position correlates only with the SYM/H index, shifting equatorward from
62° to 58° for a SYM/H increase from -40 nT to -170 nT. The lowest possible MLat is $\sim$58°.

– The equatorward shift of the WE is accpmpanied by an increase of the current density from
$I < 0.2$A/m to $\sim$1.5 A/m.

Morning sector (02–09 MLT):

The characteristics of the auroral electrojets is almost identical for midnight and morning hours.

– The MLat position at that MLT is controlled for the most part by the SYM/H activity index,
i.e., by the density of the ring current.

– In the morning sector, there exists almost exclusively the WE only.

– The current density in the WE correlates with the ASYM/H and the AL indices with maximum
values of $r = 0.69$ with respect to ASYM/H.

– The current density increases from 0.5 A/m to 2.1 A/m for intensifications of ASYM/H from
40 nT to 200 nT.

– With increasing activity, the WE shifts equatorward. The lowest observed MLat for the WE is
$\sim$58°.

The existing morphological differences between the EE and the WE probably testify differences of
the physical sources, which are responsible for the existence of the EE and WE. One possible option
is the interpretation of the EE in the evening and daytime sectors as continuation of the magneto-
spheric partial ring current (PRC) through the ionosphere via a system of FACs. The WE, which





is situated $\sim 6°$ poleward of the EE in the evening sector, might be the ionospheric continuation of the WE in the evening hours, which is connected via FACs with the central plasma sheet in the magnetospheric tail in the nighttime and morning sectors.

*Acknowledgements.* The compilation of the storms and the corresponding IMF conditions during the intervals selected was conducted by use of the one-minute OMNI data base (http://omniweb.gsfc.nasa.gov/). The
CHAMP mission was sponsored by the Space Agency of the German Aerospace Center (DLR) through funds of the Federal Ministry of Economics and Technology, following a decision of the German Federal Parliament (grant code 50EE0944). The data retrieval and operation of the CHAMP satellite by the German Space Operations Center (GSOC) of DLR is acknowledged.





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





**Appendix A  Detailed description of the dynamics of further storm intervals**

A1   The magnetic storm of 24 August 2005

This storm began with a sudden storm commencement (SSC) at 06:15 UT, which appeared as a jump-like increase of the SYM/H index up to $\sim$30 $nT$. The storm phases were identified according to the 1-min values of the SYM/H index. Fig. A1.1 shows the magnetic activity indices SYM/H, ASYM/H, AL, and the IMF components By and Bz.

The orbits 29012 and 29013 take place during the creation phase of the storm, the orbits 29014 and 29015 during the main phase, and the orbits 29016–29020 during the recovery phase. The direction and density of the Hall currents along the orbits are shown in Fig.A1.2 during daytime hours on the left hand side corresponding to the ascending orbital sections, and on the right hand side during nighttime hours for descending orbital sections. The crossings of the auroral oval occurs between

12–13 MLT during daytime and 23–24 MLT for the nighttime column. Positive values denote an eastward current (EE) for the descending orbits, and a westward current (WE) for the ascending orbital sections.

The index values during the creation phase of the storm are in the range of 25.5–32.7 $nT$ for SYM/H, 121–72 $nT$ for ASYM/H, while the substorms achieve $\sim$-1000 $nT$ according to the AL

index. SYM/H intensifies during the main phase up to -155 $nT$ and ASYM/H to 206 $nT$, where intense substorms with AL $\sim$-3000 $nT$ occur. ASYM/H values decrease to 43 $nT$ during the recovery phase, and we observe weakly variable SYM/H index values around -120 $nT$ (see Fig. A1.1).

An EE exists during daytime hours of orbit 29012 with a current density of up to 1.37 A/m at MLat 72.6°. During the subsequent orbit, the eastward current density diminishes to 0.44 A/m at MLat

70.3°. The intensification of SYM/H during the main storm phase (orbit 29015) is accompanied by a continuing decrease of the eastward Hall current to 0.38 A/m at MLat=65.4°. An EE with a density of $\sim$0.7 A/m around midday is recorded at Mlat=57.3°, i.e., below 60°, only in connection with very intense substorms (Fig. A1.1, orbit 29014). Eastward currents at such low latitudes are missing during the other orbits of this storm period. The variations of the ASYM/H index reflect

quite clearly the variations of the Hall current density: it attenuates from the orbits 29012 to 29013, and increases during orbit 29014, while it decreases again during orbit 29015.

A westward current on the daytime occurs at MLat 72°-80°, beginning with orbit 29015 and continuing until orbit 29020, i.e., throughout the recovery phase and in the absence of intense substorms. The currents achieve a maximum density of I=1.53 A/m during orbit 29018 at MLat 76.3°. This cur-

rent is controlled by IMF By>0 and changes its direction with the IMF By orientation. It is therefore definitively a PE.

The currents in the midnight sector (Fig. A1.2, right column) are generally westward directed with weak density. The only exception occurs during orbit 29014, where the current density achieves I$\sim$1.2 A/m. This orbit coincides with the development of a very intense substorm, where the Hall





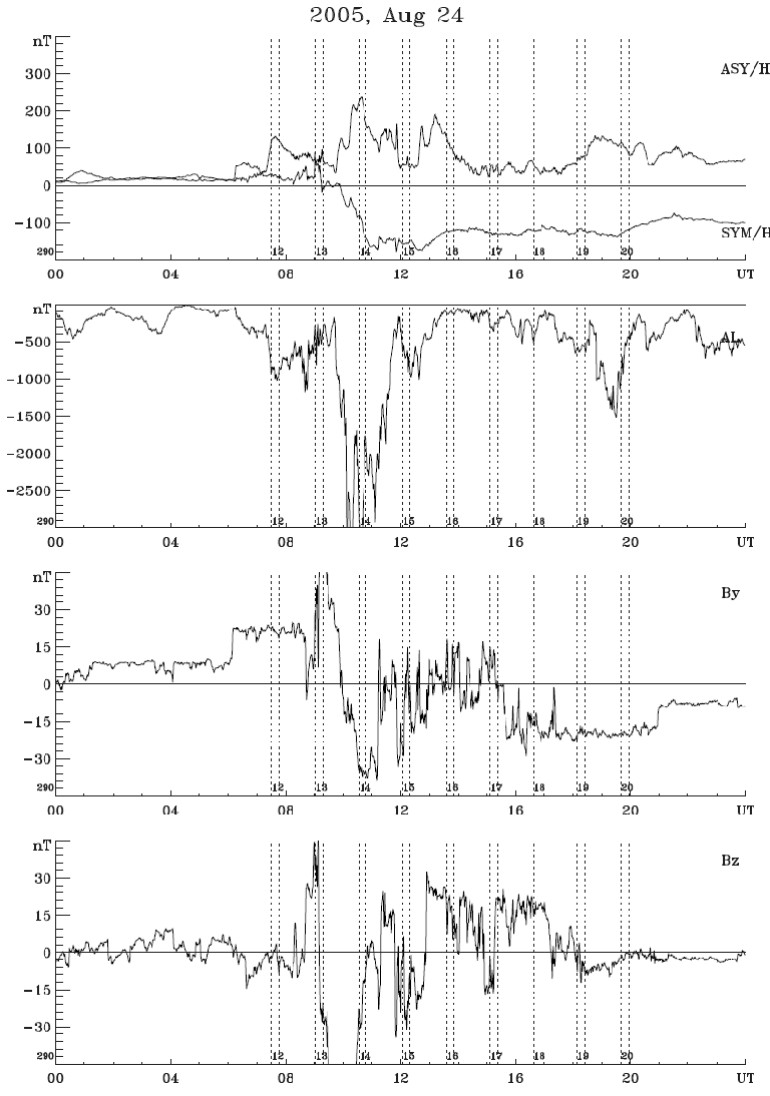

**Fig. A1.1.** One-minute values of the ASYM/H, SYM/H, and AL indices and of the By and Bz components of the IMF for the storm of 24 Aug 2005 (analysis interval from 07:00–20:00 UT, orbits 29012–29020).

current distribution is very broad with two maxima of the current density at MLat 61.2° and 73.0°. Such a broad latitudinal distribution of the auroral luminescence, with various maxima at different latitudes, is characteristic for the recovery phase of an auroral substorm (Elphinstone et al., 1996). But for the present storm of 24 August 2005, the broad splitting up in latitude appeared in the Hall currents during the main phase of the storm.





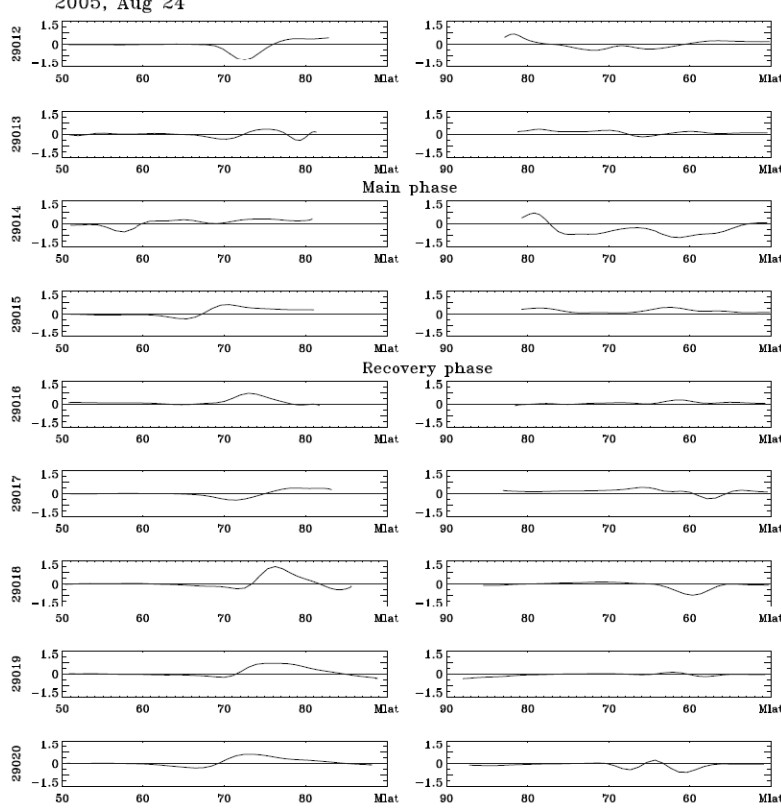

**Fig. A1.2.** Direction and density values of the Hall current along the satellite orbit at the dayside (left column, 11–13 MLT, corresponding to the ascending section of the orbit) and nightside sectors (right column, 23–24 MLT, descending orbit section). Positive currents denotes eastward current for the descending orbit section, and, accordingly, westward current for the ascending section.

Summarizing the results of Hall current observations by the CHAMP satellite during the magnetic disturbance period of 24 August 2005 in the daytime and nighttime sectors (11–13 MLT and 23–24 MLT, respectively) we conclude:

For the midday sector:

– An EE with a current density of 1.37 A/m exist during the creation phase at MLat$\sim$73.0° for

830        substorms in the auroral zone with intensities of AL$\sim$-1000 nT.

– The EE is observed at MLat<60° during the main storm phase for intense substorms with intensities of AL$\sim$-3000 nT.

– The variations of the EE intensities during the creation and main phases of the storm occur





synchronous with the ASYM/H index. Comparable variations with the SYM/H index are not

observed.

   – Westward or eastward directed currents are observed during the recovery phase at 72°<MLAT<80°
      with a maximum density of ∼0.9 A/m. Their direction is controlled by the IMF By compo-
      nent, i.e., they are in accordance with the PE.

For the midnight sector:

– As a rule, the Hall currents are westward directed during nighttime. In the concrete observa-
      tions, the WE can be splitted into several parts with several maxima versus latitude.

### A2 The magnetic storm of 18 June 2003

Fig. A2.1 shows the variations of the SYM/H and ASYM/H indices for the magnetic storm of June
18th, 2003. The storm phases are represented by the orbit numbers 16532 and 16533 for the creation

phase, 16534–16536 for the main phase, and 16537–16541 for the recovery phase. Extreme values
of SYM/H and ASYM/H are observed during the main phase with -163 $nT$ and 91 $nT$, respectively,
while the substorm index AL achieves -1298 $nT$. THE CHAMP trajectories are situated during this
storm period along the meridional plane of 13–14 MLT (afternoon) and 00–02 MLT (near midnight).
In the daytime sector, a EE exist during the creation phase at MLat ∼67° with I∼0.43 A/m, and a

WE at MLat ∼72° with I∼0.42 A/m. Both electrojets are retained during the main storm phase with
an EE of I∼0.8 A/m at MLat ∼62° and a WE of I∼0.5 A/m at MLat ∼67°. The WE only persists
during the recovery phase with I∼0.3 A/m at MLat ∼78°(orbits 16537 and 16538) This high-latitude
westward current near MLat ∼77°with I∼0.4 A/m does not vanish till the end of the recovery phase.
Such a high-latitude position of a westward current near noontime MLT gives reason to suggest that

this is a polar electrojet (PE). This assumption would apply, if the IMF By component is negative.
Indeed, the By component appeared to be at a steady neagative value during the orbits 16537–16541.

    As a rule, the ionospheric currents in the nighttime sector are westward directed in the MLat range
of 57.8°–63.0° with I∼0.5 A/m. Only during two orbits in the creation and main phases, the current
density achieved I∼(1.1–1.4) A/m.

It should be noted that this storm had relatively intense SYM/H values, while the ASYM/H values
remained however at a relatively low level. The EE and WE intensities were small as well.

    Summarizing the results of Hall current observations by the CHAMP satellite during the magnetic
disturbance period of 18 June 2003 in the daytime and nighttime sectors (13–14 MLT and 00–
02 MLT, respectively) we conclude:

– The quite strong geomagnetic storm (according to the SYM/H<-150 nT index value during
      the main phase) is accompanied by substorms with AL up to -1500 nT and with the lowermost
      index value for the asymmetry of the field ASYM/H<-100 nT. The peculiarities of this storm



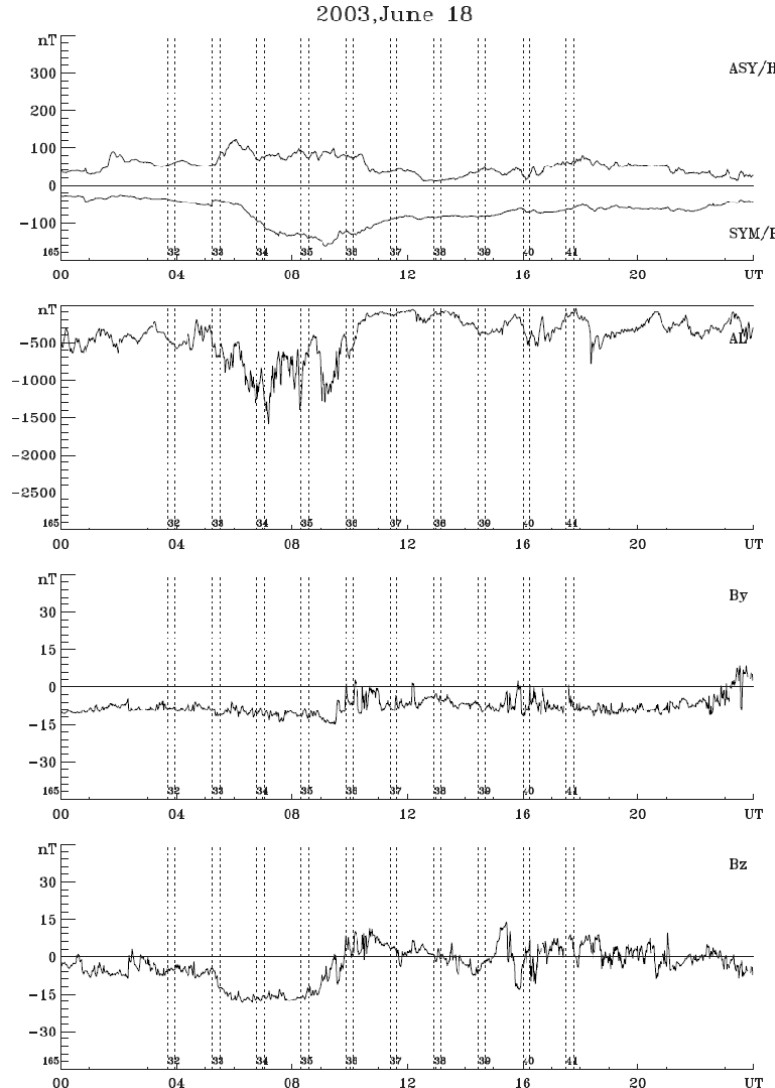

**Fig. A2.1.** One-minute values of the ASYM/H, SYM/H, and AL indices and of the By and Bz components of the IMF for the storm of 18 June 2003 (analysed interval from 03:00–18:00 UT, orbits 16532–16541).

period caused obviously the appearance of an EE in the daytime sector and a WE in the nighttime sector at MLat<60°.

– A stable PE with a current density up to 0.4 A/m in westward direction persists during the recovery phase with an IMF By<0 nT component.




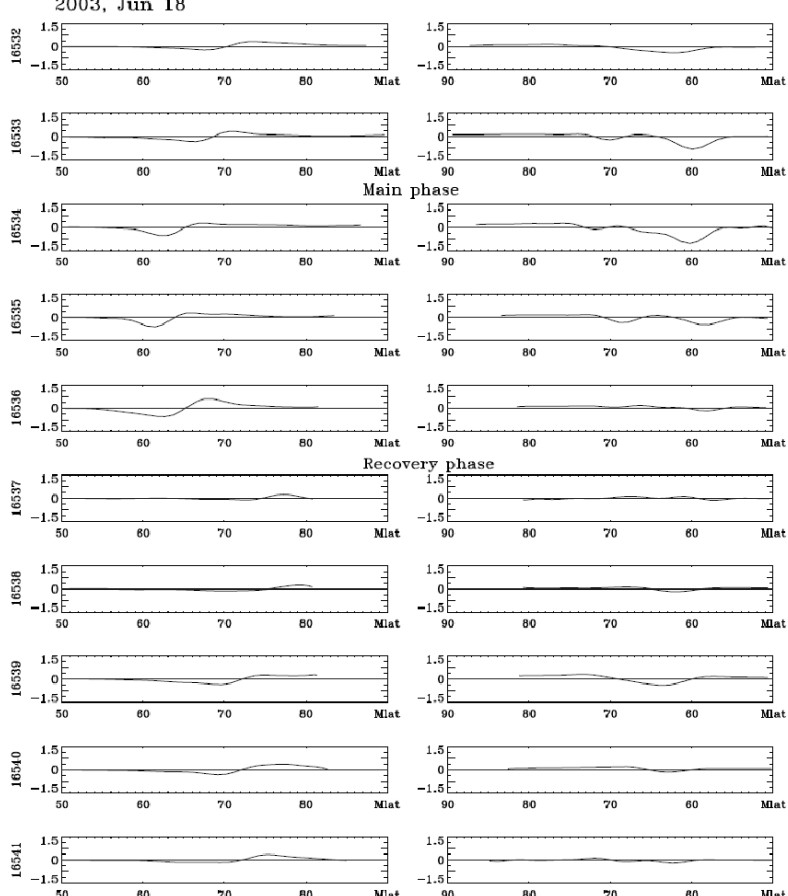

**Fig. A2.2.** Direction and density values of the Hall current along the satellite orbit at the dayside (left column, 12–16 MLT, corresponding to the ascending section of the orbit) and nightside sectors (right column, 00–04 MLT, descending orbit section). Positive currents denotes eastward current for the descending orbit section, and, accordingly, westward current for the ascending section.

A3  The magnetic storm of 30 May 2005

Fig. A3.1 shows the variations of the SYM/H, ASYM/H, and AL indices for the magnetic storm of May 30, 2005, between 02 UT and 20 UT. The vertical dotted lines indicate the time intervals of the satellite crossings over high latitudes of the Northern Hemisphere (MLat>60°), and the numbers denote the satellite's orbit counter. Prior to the storm onset (orbits 27659 and 27660), the geomagnetic field is according to all indices, including the AL index, relatively quiet. It is recovered from -28 $nT$ to -17 $nT$ in terms of SYM/H, from -38 $nT$ to -18 $nT$ for ASYM/H, and from -40 $nT$ to 0 $nT$ with





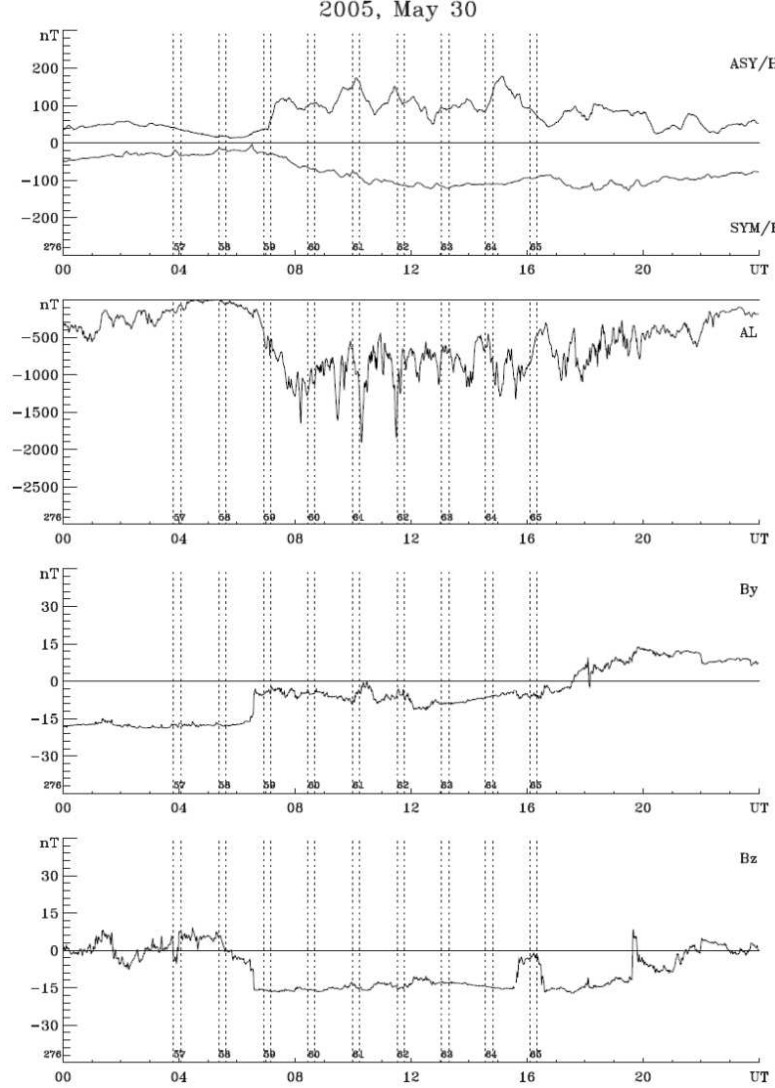

**Fig. A3.1.** One-minute values of the ASYM/H, SYM/H, and AL indices and of the By and Bz components of the IMF for the storm of 30 May 2005 (analysis interval 02:00–17:00 UT, orbits 27658–27667). The time of each orbit and its orbit number are indicated as in Fig. A1.1.

respect to the AL index. These changes correspond to a recovery process toward a quiet time level

after the previous disturbance.

The main phase of the magnetic storm starts with a steady increase of SYM/H from -29 $nT$ during

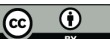



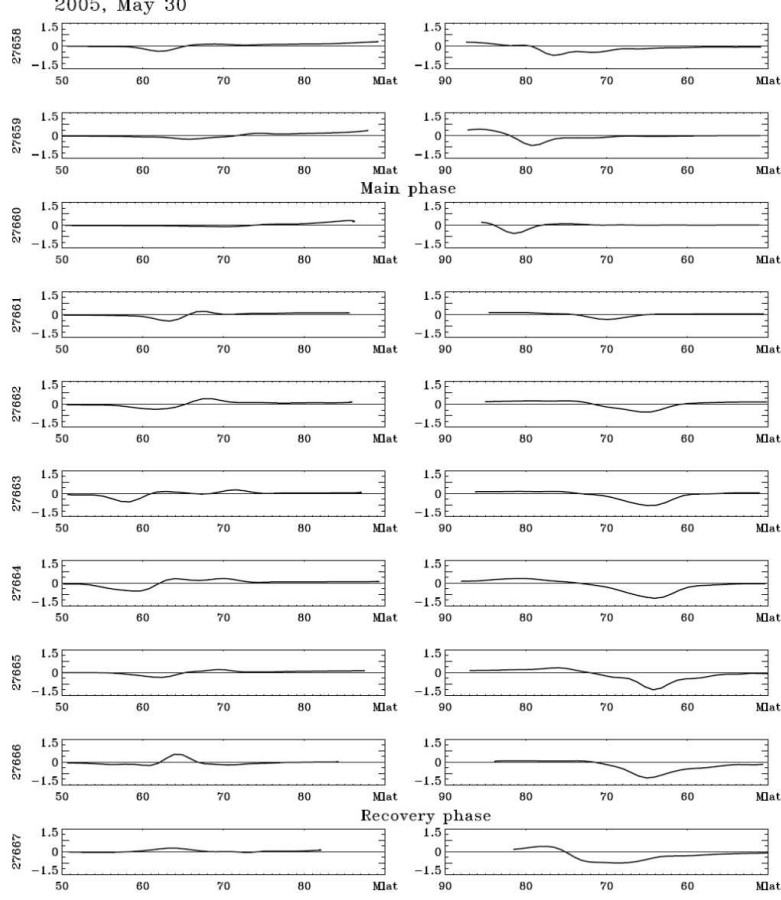

**Fig. A3.2.** Direction and density values of the Hall current along the satellite orbit at the duskside (left column, 19–21 MLT, corresponding to the ascending section of the orbit) and dawnside sectors (right column, 06–09 MLT, descending orbit section). Positive currents denote an eastward current flow for the descending orbit section, and, accordingly, westward current for the ascending section.

orbit 27661, a jump-like increase from 44 $nT$ to 104 $nT$ in ASYM/H during the same overflight and continues with an increase of SYM/H to -118 $nT$ during orbit 27665. The peak values of ASYM/H and AL during the main storm phase are 162 $nT$ and -1200 $nT$, respectively. The recovery phase takes place during the orbits 27666 and 27667, after which during the orbit 27668 the appearance of a new disturbance is recorded (according to the AL and SYM/H indices). The ascending CHAMP trajectory during the storm goes along the 19–21 MLT meridian (evening), while the descending orbit section is along the 06–09 MLT meridian in the morning sector. Fig. A3.2 shows the direction



and the density of the Hall currents for the evening (left side) and morning (right side) sectors.

During the orbits prior to the beginning of the main phase, the Hall current is either missing in the evening sector or exists only in terms of a distributed eastward current with maximum densities of $J{\sim}0.3$ A/m at MLat${\sim}66°$. In the morning sector, a WE is recorded with $J{\sim}0.9$ A/m at MLat${\sim}80°$ and MLT${\sim}09$ hours. The existence of such intense currents during daytime hours at such high latitudes during relatively quiet geomagnetic conditions is unusual. A reasonable explanation might be the

assumption that this current concerns the PE. In this case, the orbits investigated should occur during conditions of IMF By$<0$ $nT$. Indeed, according to Fig. A3.1 a quite stable negative IMF By${\sim}$-18 $nT$ is observerd prior to the main storm phase The beginning of the main phase (orbit 27661) is characterized by the appearance of two currents in the evening sector: the EE with $J{\sim}0.6$ A/m at MLat${\sim}63°$ and the WE with $J{\sim}0.3$ A/m at MLat${\sim}68°$. In the course of the storm, the EE attains

a density of $J{\sim}0.7$ A/m, shifting equatorward until MLat${\sim}80°$. The displacement in MLat toward the equator reflects the more general tendency, according to which the the electrojets move more equatorward with increasing current $J$. The current density in the WE retains at $J{\sim}0.4$ A/m. In the morning sector, the current stays at MLat${\sim}70°$, and its current density during orbit 27661 is kept at $J{\sim}0.4$ A/m. This is obviously the first appearance of an auroral WE in the morning sector. The WE

at auroral latitudes increases during the subsequent orbits and attains 1.5 A/m during orbit 27665 at MLat${\sim}64°$. The recovery phase during orbit 27667 is characterized by a westward current with $J{\sim}$-1.0 A/m in the morning sector at MLat${\sim}68°$ and a weaker current with $J{\sim}0.3$ A/m at MLat${\sim}63°$,. In the course of the storm, the current density $J$ in the morning sector exceeds significantly the Hall current density values of the same orbit in the evening sector.

Summarizing the results of Hall current observations by the CHAMP satellite during the magnetic disturbance period of 30 May 2005 in the dusk and dawn sectors (19–21 MLT and 06–09 MLT, respectively) we conclude:

– Two auroral Hall currents (EE and WE) exist in the evening, and one current only (WE) in the morning sector;

– The currents are positioned, as a rule, at latitudes MLat of the auroral zone ($63°$-$68°$). During the main phase, the current can be shifted to MLat${\sim}58.5°$;

– In the evening sector, the position of the EE is more equatorward than the WE;

– During early evening hours, the Hall current density of the EE exceeds the WE current density, and in the morning hours the WE current density is larger than during the evening;

– The recovery process toward the quiet-time level can be accompanied at by late evening or polar electrojet (PE) at MLat${\sim}80°$ in the late morning hours of the PE with $J{\sim}0.8$ A/m.





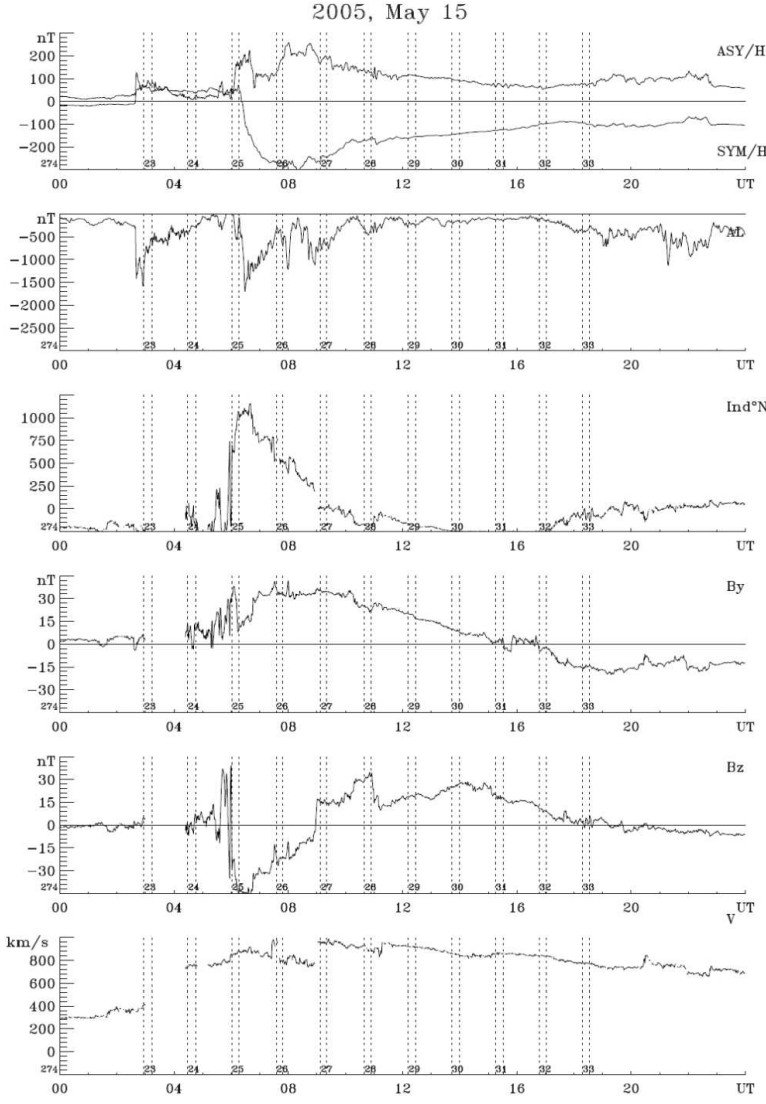

**Fig. A4.1.** One-minute values of the ASYM/H, SYM/H, and AL indices and of the By and Bz components of the IMF for the storm of 15 May 2005 (analysis interval 00:00–19:00 UT, orbits 27423–27432). The time of each orbit and its orbit number are indicated as in Fig. A1.1.

A4    The magnetic storm of 15 May 2005

Fig. A4.1 shows the variations of the SYM/H, ASYM/H, and IndN indices, as well as the IMF By and Bz components and the solar wind velocity in the interval 00–23 UT for the magnetic storm of May



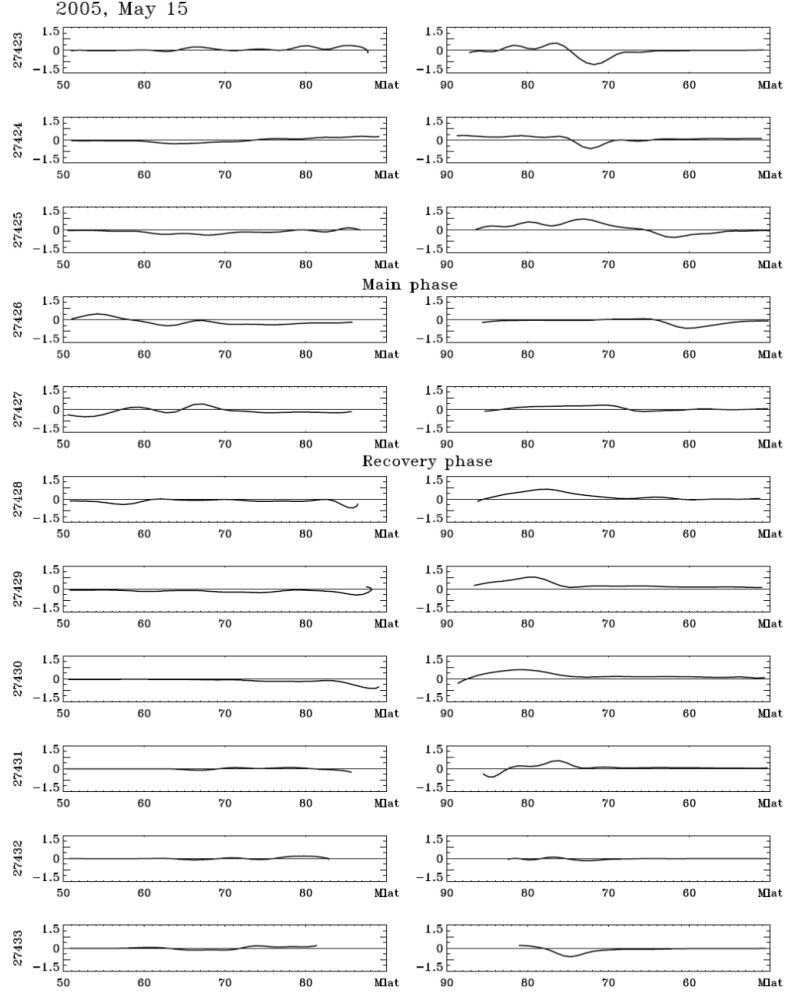

**Fig. A4.2.** Direction and density values of the Hall current along the satellite orbit at the duskside (left column, 20–22 MLT, corresponding to the ascending section of the orbit) and dawnside sectors (right column, 08–10 MLT, descending orbit section). Positive currents denote an eastward current flow for the descending orbit section, and, accordingly, westward current for the ascending section.

15, 2005. The main phase of the magnetic storm takes place during the orbits 27426 and 27427 with a SYM/H index value of $\sim$-274 $nT$, an ASYM/H of $\sim$186 $nT$, and AL $sim$-1700 $nT$. The orbits 27423 and 27424 prior to the main phase occur during weakly disturbed magnetic field conditions with SYM/H $\sim$50 $nT$ and ASYM/H $\sim$75–16 $nT$. During orbit 27425 with SYM/H $\sim$48 $nT$, the ASYM/H index increases strongly to $\sim$121 $nT$, which appears to be the onset of an intense magnetic



storm. The recovery phase takes place during the orbits 27428 and 27432, during which occurs a
steady decrease of the SYM/H index to $\sim$-125 $nT$ and of the ASYM/H index value to $\sim$70 $nT$.
The ascending CHAMP trajectory during the storm spread along the 20–22 MLT meridian, and the
descending trajectory is along the 08–10 MLT meridian in the morning sector. Fig. A4.2 shows the
direction and the density of the Hall currents for the evening (left side) and morning (right side)
sectors.

The current density for the EE during orbit 27425 is with $\sim$0.4 A/m quite small in the evening
sector prior to the main phase. Both an EE and a WE exist during the main phase with J$\sim$0.6 A/m.
The EE shifts on average to a MLat of $\sim$52.5° with SYM/H $\sim$-250 $nT$. The Hall currents are
practically absent during the recovery phase.

In the evening sector, the currents turn out to have difficult characteristics, changing with the storm
phases. A WE at MLat$\sim$72° with J$\sim$1.0 A/m is recorded during the magnetically quiet period prior
to the main phase. With the development of the main phase, the WE shifts to MLat$\sim$61°. During
the recovery phase, the WE decays at auroral latitudes, but in the latitudinal range 77°<Mlat<80°
an EE appears with J$\sim$1.0 A/m (orbits 27428–27431). The orbits with an EE coincide temporally
with an interval of IMF By>0 $nT$ in the solar wind (Fig. A4.1). All characteristic features of the PE
are therefore present here. During orbit 27433, the direction of the current changes to WE. This is
accompanied by a corresponding change of the IMF By orientation as can be seen in Fig. A4.1.

The characteristic features of this storm are the following:

– The quiet-time level of the magnetic field variations prior to the storm main phase can be
describes as missing or unimportant intensities of the EE and WE Hall currents in the evening
    sector, while in the morning sector exists only the WE at auroral latitudes.

– During the main phase of this intense storm with a SYM/H index of $\sim$-250 $nT$ in the evening
    sector, the WE shifts to MLat$\sim$52.5°, and the WE to MLat$\sim$54.0°.

– A PE appears during the recovery phase in the late morning hours at 77°<Mlat<80°, where
the Hall currents are controlled by the direction of the IMF By component.

A5   The magnetic storm of 18 August 2003

Fig. A5.1 shows the variations of the SYM/H, ASYM/H, and AL indices, as well as the IMF By
and Bz components in the interval 00–23 UT for the magnetic storm of August 18, 2003. The main
phase of the magnetic storm takes place during the orbits 17482–17489 with peak values of SYM/H
and ASYM/H of $\sim$-135 $nT$ and $\sim$101 $nT$, respectively, and an AL value of $sim$-1400 $nT$. During
the orbits 17480 and 17481 prior to the main phase the values of SYM/H and ASYM/H are $\sim$-18 $nT$
to -43 $nT$ and $\sim$72–49 $nT$, and during the recovery phase in the course of orbits 17490–17493 they
amount to $\sim$-115 $nT$ and $\sim$56 $nT$, respectively. The CHAMP trajectories during the storm spread
along the 07–09 MLT meridian (morning) and along the 19–21 MLT meridian(evening). Fig. A5.2




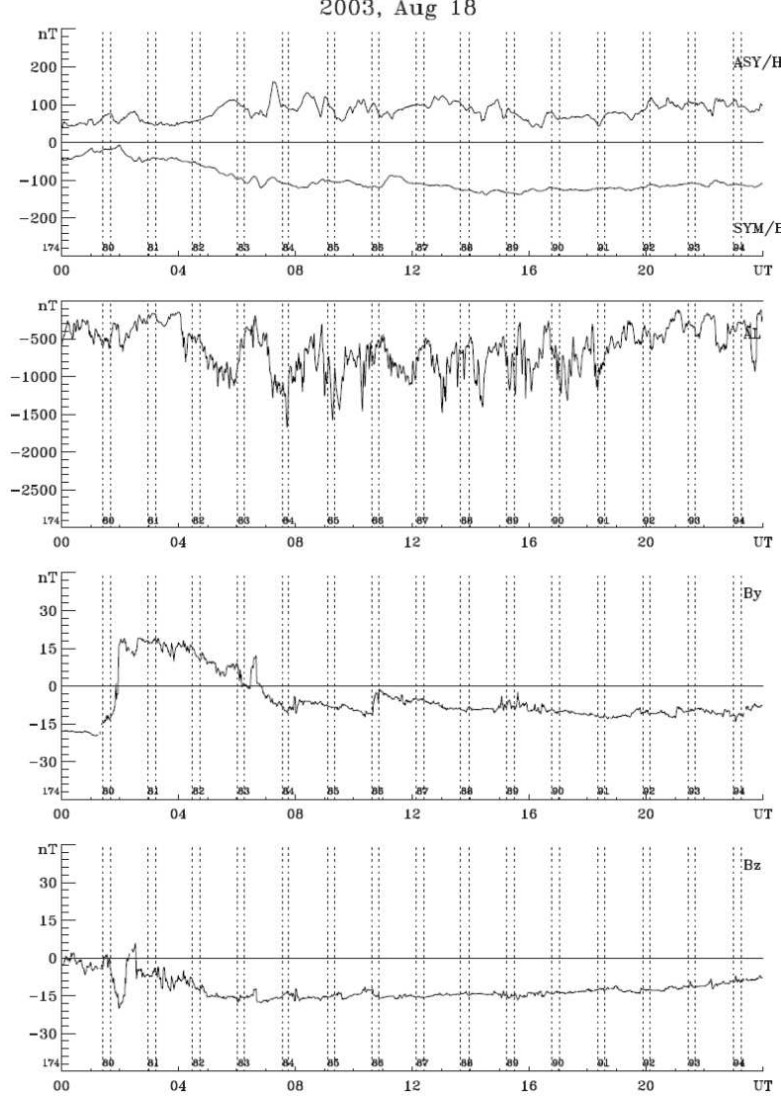

**Fig. A5.1.** One-minute values of the ASYM/H, SYM/H, and AL indices and of the By and Bz components of the IMF for the storm of 18 Aug 2003 (analysis interval 00:00–23:00 UT, orbits 17480–17494). The time of each orbit and its orbit number are indicated as in Fig. A1.1.

shows the direction and the density of the Hall currents for the morning (left side) and evening (right side) sectors.

The characteristic peculiarities of the spatial-temporal distribution of the FACs during this storm





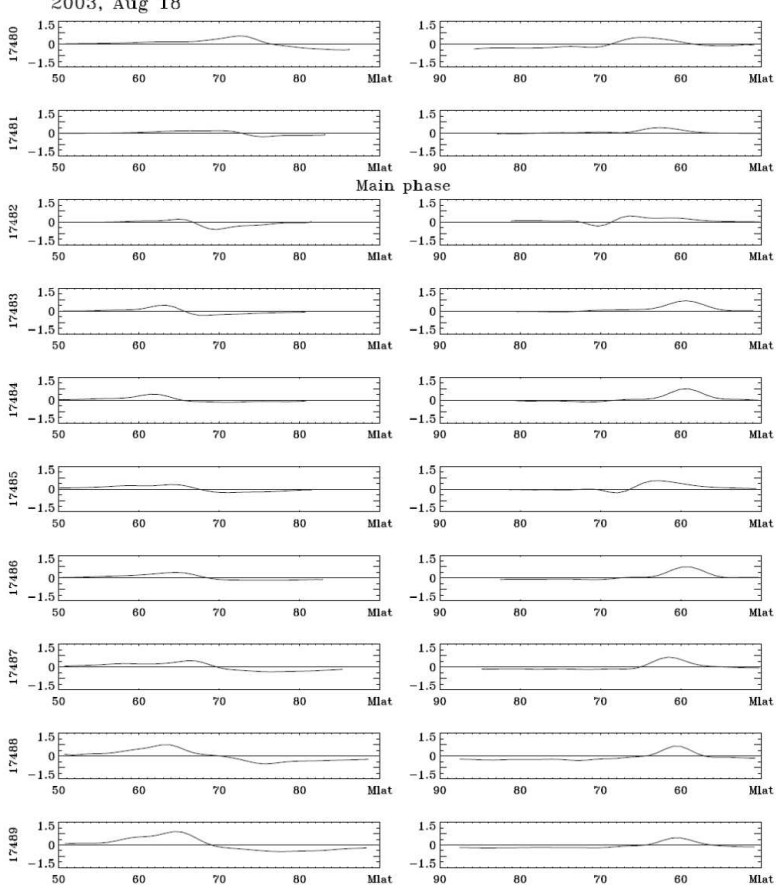

**Fig. A5.2.** Direction and density values of the Hall current along the satellite orbit at the dawnside (left column, 07–09 MLT, corresponding to the ascending section of the orbit) and duskside sectors (right column, 19–21 MLT, descending orbit section). Positive currents denote an eastward current flow for the descending orbit section, and, accordingly, westward current for the ascending section.

concur with those described for the other storms. During the main phase in the evening sector, there exists, as a rule, an EE. The EE appears at MLat$\sim$66.5° with J$\sim$0.6 A/m, and shifts then equatorward to MLat$\sim$58.8° with J$\sim$1.0 A/m during orbit 17486. A WE exists in the morning sector at auroral latitudes of 61°<Mlat<65° with J$\sim$1.2 A/m. A weak distributed eastward current in the polar cap persists due to the closure of parts of the electrojets across the near-polar region.