# Peer review of "Dynamics of the electrojet during intense magnetic disturbances"

_Annales Geophysicae, 2018_

## Referee Comment (RC1) · Anonymous Referee #1 · 4 Jun 2018

**General comments**

The manuscript analyze high latitude ionospheric Hall currents derived from scalar magnetometer meaurements from LEO orbit from the CHAMP satellite during 6 different geomagnetic storm periods. On the basis of these data, the estimated Hall electrojets are characterized in different MLT sectors, and interpreted as either the Eastward (EE), Westward (WE), or Poleward electrojet (PE), the latter occurring only on the dayside in vicinity of the cusp, also in an east/west direction.

The manuscript serve to characterize how the EE, WE, and PE vary in location (latitude) and strength in the different MLT sectors during the different strom phases. Possible relationships to geomagnetic and IMF/SW indices are explored through a regression and correlation analysis.

[Figure]

The results appear reasonable, however, the manuscript presently fails to place the results into the present state of knowledge in the field. One example is that much attention is focused on the latitudinal variability of the EE and WE, where the electrojets are mainly found to move equatorward in the storm main phase, compared to the initial and recovery phase. However, this result in not discussed together with the expanding/contracting nature of the open magnetosphere (e.g. Cowley and Lockwood 1992, Milan 2015), largely explaining why. This needs to be mentioned, and I think the manuscript would benefit from more rather focusing on the specific latitudes that the expansion reach (that they also mention), with relation to e.g. the SYM/H index and the Newell coupling function.

Again, for the discussion section, I would very much appreciate to see how the present results add to the earlier works cited in the introduction. Presently the discussion is largely void of references, and it is not at all clear what the present manuscript has specifically done that is different from the earlier works, other than the method.

Cowley, S. W. H., Lockwood, M. (1992). Excitation and decay of solar wind-driven flows in the magnetosphere-ionophere system. Annales Geophysicae, 10(1–2), 103–115.

Milan, S. E. (2015). Sun et Lumière: Solar Wind-Magnetosphere Coupling as Deduced from Ionospheric Flows and Polar Auroras. In S. Cowley FRS, Stanley W. H. and Southwood, David and Mitton (Ed.), Magnetospheric Plasma Physics: The Impact of Jim Dungey's Research (Vol. 41, pp. 33–64). Springer International Publishing. https://doi.org/10.1007/978-3-319-18359-6

**Specific comments**

Title: I find the word "Characteristics" more suitable than "Dynamics" for the title. As the manuscript focus on the correlation of the location and strength of the electrojets deduced from a LEO satellite, with geomagnetic and SW/IMF variables, the study is more characterizing in nature, than describing dynamics, which is usually referred to as changes on shorter time scales than 90 min.

Abstract: The extensive listing is unusual. The abstract might be too long for the journal. I would prefer a shorter summary of the method and findings.

Introduction The introduction mention quite a few historic works on the characterization of ionospheric horizontal currents at high latitudes. Although this is highly relevant for the present manuscript, the authors some times fail to convey the implication of the work they describe. The introduction often reads more like a brief summary of the previous works, mainly mentioning what they have been measuring, and not putting it into broader context. In particular this is evident in the paragraph starting on line 98. I would suggest to remove this paragraph unless the authors can more directly state how this is relevant for the present study. I also have a similar concern with the paragraph starting at line 160. This is mostly a listing of what the referenced literature observe, no explanation of their result is provided. For the reader it would be very satisfactory to briefly explain their findings. Example: Why does the electrojets expand to lower latitudes during increased activity (the oval expands...)? Also, how the electrojets disrupt into space could be elaborated more on, if not removed. The same arguments goes for the paragraph starting at line 176. Please provide how their findings are relevant for the present manuscript. At the moment it only states that Wang et al investigated electrojet currents from space, which is not very informing.

Lines 96-97: "Potential differences at cusp latitudes". In the present manuscript cusp latitudes is refereed to as a point. To get a potential difference, a distance need to be involved. It is not obvious what is the distance in this case. I think you refer to the

distance where dayside merging occur. To get around this, maybe an electric field, or velocity value in the east/west direction could be used?

Lines 134-159: Suggest to merge this into one paragraph as both paragraphs are dealing with the same Ritter et al 2004 paper.

Data: Line 215: Need to define $\Phi$ (or $\Phi_{PC}$ as Newell calls it): Magnetic flux inside polar cap. The coupling function describe the change of this related to reconnection at the magnetopause, $d\Phi_{MP}/dt$. Together with the nightside reconnection rate ($d\Phi_N/dt$), this control the expansion/contraction of the polar cap ($d\Phi_{PC}/dt$). This is relevant for the present study as expansion/contractions of the polar cap lead to similar displacements of the electrojets. [Cowlet and Lockwood, 1992].

Also relevant for this study is the time averaging of the solar wind data used to estimate IndN. Using a long interval (20-60 min) would likely produce a better correlation to e.g. AL, than using the instant 1 min values. How is this done in the present manuscript?

Method: A short introduction to the ionospheric currents at high latitudes would be appreciated, along the lines that we can decompose the currents into Hall, Pedersen, and Birkeland currents, all contributing to the field measured at LEO altitude, but mainly the Hall currents are detected on the ground as explained by Fukushima (1976). As no information regarding the electric field is provided in the manuscript (needed to define the directions of Hall and Pedersen currents), it should state something like "we assume Hall currents to be divergence free, i.e close entirely within the ionosphere, while the Pedersen currents are curl free." Laundal et al. 2018 showed (see their Figure 14) that during summer conditions, which is the case in for the present manuscript, the divergence free and curl free ionospheric currents mainly represented the Hall and Pedersen currents, respectively.

Lines 241-242: Suggest that you write that only one of the events in Table 1 is presented in the paper, and the rest are included in the Appendix. (a rewrite of the present sentence)

Line 248: which is the symmetric part of the ring current. As you write on line 252, the ring current is in general asymmetric.

Lines 255-260: I think these sentences belong to the introduction. Is this connection of the EE to the PRC the same as described in lines 166-168? If so, it could be moved into that location to give better background information.

Fukushima, N.: Generalized theorem for no ground magnetic effect of vertical currents connected with Pedersen currents in the uniform-conductivity ionosphere, Rept. Ionos. Space Res. Japan, 30, 35–40, 1976

Laundal, K. M., Finlay, C. C., Olsen, N., Reistad, J. P. (2018). Solar wind and seasonal influence on ionospheric currents from Swarm and CHAMP measurements. Journal of Geophysical Research: Space Physics, 123. https://doi.org/10.1029/2018JA025387

Events: Line 265: Is this number referring to the SYM/H index? That should be explained before presenting the values. Also a reference to the figure showing SYM/H and the orbits should appear here. Further, from looking at the SYM/H data at the WDC Kyoto webpage, it is provided as integer numbers. How does the decimal number arise?

Figure 1 is never introduced. This need to be improved. It would also be of interest to show the AU index in this figure as the paper discusses measurements of the EE during this interval.

Figure 2: No label or units on the y-axes. Accordingly → respectively, and put to the end of the sentence.

Lines 282-286: It is not obvious if the method allows for the determination of the direction of the Hall currents. Is an east/west direction assumed? Sign information is provided in Figure 2, however, the text does not explain what direction positive currents represent (only the figure caption does).

Section 4.1: Although this subsection is devoted to show variations of the observed Hall

currents with relation to the SYM/H index, the EE and WE is much discussed. I find it therefore appropriate to mention the expected closer relationship of the observations to the AU and AL index, which is regarded as a measure of the auroral electrojet from the ground.

Lines 378-380: Despite showing the AL index, the authors have not discussed the trend of the measurements with respect to this index.

Lines380-381: For the reader to judge this statement, a measure of the dependency should be provided, e.g. a correlation coefficient. Although mentioned in the text, this dependency is really hard to see from looking at Figures 1 and 2. This become much more evindent in the later figures where the correlation analysis is performed. This conclusion should therefore not be made at this point.

Section 4.2: From line 363 the manuscript summarizes the observed trends. This should therefore not be a part of subsection 4.2, but rater ha separate subsection (if the summary is really needed), or all the subsections could be combined into one section.

Discussion: The present discussion does a poor job in relating the observational findings (which is the majority of what is presented in this section) to the state of the art knowledge of the characteristics of the high latitude electrojets. Are the present findings mostly in agreement with the present knowledge, and where does it represent new findings/analysis that could lead to new knowledge? To my knowledge, a possible link between the asymmetric ring current and IMF By has not been established earlier. That is one piece of information that I think the present manuscript is hinting towards.

Figure 3: The caption should indicate what the red and blue points refer to.

Lines 437-439: If the PE variations during quiet intervals are not due to IMF By, what could it be then? I don't understand this sentence.

Lines 452-489: In my opinion, the term "solar wind electric field penetration" is a misleading term/concept. My take on this is that reconnection opens field-lines on the day-side. During By conditions, these newly opened field lines has a strong bend leading to tension force in the east/west direction. Field-aligned currents facilitate the energy needed to reconfigure these field lines (there will be a drag in the ionsphere due to collissions), and associated east/west plasma flows are seen in the ionosphere, together with the field aligned currents and their associated horizontal currents in the ionosphere, who's magnitude is determined by the ionospheric conductivity. On the ground one observe mainly (at least during summer) the divergence free Hall currents related to this, your PE. In this description it becomes meaningless (in my opinion) to say that the solar wind electric field (originating from the Lorentz transformation) penetrate into the high latitude ionosphere to drive currents. Hence, in order to foster understanding, to distinguish cause and effect, being one of the most fundamental tasks for physicists, I think the "electric field penetration" is a bad term for this application. That being said, I don't see how these paragraphs really fits into the context of characterizing the PE. I therefore suggest to remove these paragraphs.

Lines 495-499: This result has the strongest correlation. Could the authors try to elaborate on possible explanations for this result? The ASYM/H describe to some extent an asymmetry in the inner magnetosphere. The PE arises, as the manuscript shows, due to IMF By, which is maybe the most important paramater affecting the asymmetry in the magnetosphere. From simple linear correlation of |IMF By| and ASYM/H one get r=0.29, while with SYM/H one get r=-0.16 (using 5 min OMNI data from 1981-2016). This supports the results in the manuscript, and suggests that IMF By could be a source of asymmetry in the ring current.

Line 540: What is meant by "changing SYM/H" index? Is there a criterion in the data selection that the SYM/H should drop by e.g. a fixed amount? If so, how what is the reason for that?

Line 551: Strictly speaking, IndN does not characterize a current function, but is designed to be proportional to the amount of opened magnetic flux per unit time, which

in turn affect the current systems.

Line 563: Need to introduce Figure 5.

Line 282-583: The mentioned trend of Mlat vs IndN is not obvious from Figure 6d.

Lines 595-596: I disagree. As they should in principle be the same, the large observed spread (Figure 7c) give insight to the difference in technique used to probe the WE (CHAMP and AL index). I was surprised the relationship was not clearer. Could some of this variation be due to difference is in sampling/averaging window?

Conclusion: Similar to my comment regarding the abstract, I find the conclusion section very long. It is a very comprehensive listing of the detailed discussion provided earlier. I would have appreciated if the authors would have rather summarized what they believe is the most important and significant among their findings.

Lines 606-712: This has not been mentioned before and should therefore be moved to the discussion

**Minor comments**

Line 10: IndN needs to be defines. Suggest that you only refer to it as the Newell et al coupling function in the abstract.

Line 24: Should this be 80 degrees MLAT, to be consistent with the equatorward motion mentioned?

Line 28: Need to define acronyms in abstract. Or maybe just refer to it as ring current

in the abstract.

Lines 74-77: What level of confidence does this refer to? Could a reference or a name of the formula be provided?

Line 111: compilation → comparison

Line 112: demonstrates

Line 131-132: From looking at Figure 26 in Sandholt et al 2004, theese fast convection channels are located on open field lines, hence poleward of the OCB and not at the OCB as the manuscript states.

Lines 134, 139, and 153, and throughout the text: The use of "density" refering to the strength of the electrojets is sometimes unfortunate in my opinion. It can easily me mixed with how dense the current lines in the model is placed. Consider to change it to "strength".

Line 151: Delete "currents"

Line 160: EISCAT is an incoherent scatter radar network, not a magnetomenter network

Line 180: electromagnetic

Line 207: 12 observatories

Line 219: BT is actually only constricted from the y and z components of IMF

Line 220: One should rather use theta = arctan2(By,Bz), which is a common algorithm that handle the different signs of By and Bz properly, compared to the mathematical function arctan().

Line 233: delete "there"

Line 236: density → level

Line 245: Suggest to delete "mean"

Line 282: density → intensity

Line 331: delete "there"

Line 395: Transferred is not a good word here.

Lines 405-406: Suggest to replace "occurrence and dynamics" with "strength" and "domains" with "processes"

Line 409: Delete "from geomagnetic activity researches"

Lines 411-412: This sentece is not related to any of the following text.

Line 415: add "all the" before 6

Line 414: Add something like "versus IMF and geomagnetic indices"

Lines 419-420: I dont understand this sentence.

Line 511: Suggest to rewrite: "... correlation of IndN with PE Mlat (r=0.52)."

Lines 517-518: Suggest to delete this sentence.

Line 528: Suggest to add something like "from the individual orbits". As I read these figures, each dot correspond to a specific orbit within the MLT range, among the 6 events studied.

---

## Referee Comment (RC2) · Anonymous Referee #2 · 26 Jun 2018

Review of Manuscript angeo-2018-31: "Dynamics of the electrojet during intense magnetic disturbances" by L. I. Gromova et al.

In the manuscript the authors analyze total magnetic field data from the CHAMP satellite during six periods of magnetic storms with respect to the relation between the phases of the storms and those of the polar and auroral electrojets. A somewhat complex procedure allows deriving sheet current densities of the Hall currents flowing underneath the spacecraft. The measurements were made during Northern summer periods and stem from noon-midnight as well as dawn-dusk crossings of the spacecraft. The main products consist of correlations between the measured Hall current locations and strengths with auroral and ring current activity as quantified by various geomagnetic indices. The results allow some interesting insights into the mutual dependences or lack of them.

The presented text contains some valuable data deserving publication. However, the presentation is too detailed and tedious to read. The main products are well presented in Figures 4 to 7 and their discussion appropriate. However, the conclusions are not concise, but rather repetitive of the earlier description of the data. One misses interpretations in terms of the build-up of the asymmetric and symmetric ring currents and the role played in that by the high-latitude magnetosphere as indicated by the behavior of the polar and auroral electrojets. I recommend to encourage the authors to shorten strongly Abstract, Introduction, and Conclusions and concentrate in the latter on the meaning of the most striking correlations.

Two additional comments.

The polar and auroral electrojets are treated as if there was no relation. One is missing a reference to the Region 1 and 2 field-aligned current systems.

Pages 17 and 18 contain estimates of electric field penetration from the solar wind into the magnetosphere. The authors seem to take this concept literally. There is no direct electric field penetration. There is reconnection and maybe some viscous interaction by plasma waves between the solar wind and the Earth's magnetic field. Thereby forces are being transferred via magnetic shear stresses. The direction of By matters of course. Electric fields inside the magnetosphere stem from the application of these stresses to the ionospheric plasma. Therefore, the calculated percentages of electric field transfer from a partial magnetic field component are meaningless.

---

## Author Comment (AC1) · 22 Aug 2018

Please find attached our responses to the review of Referee 1 as well as the revised manuscript and an annotated manuscript, which indicates all the changes/additions in red.

Please also note the supplement to this comment:
https://www.ann-geophys-discuss.net/angeo-2018-31/angeo-2018-31-AC1-supplement.zip

---

## Author Comment (AC2) · 22 Aug 2018

Attached please find the zip file, which contains our answers to the reviews of both Referees, the revised manuscript, and the annotated manuscript with all changes/additions in red.

Please also note the supplement to this comment: https://www.ann-geophys-discuss.net/angeo-2018-31/angeo-2018-31-AC2-supplement.zip
* * *

---

## Author Response (AR1)

**General comments**

The manuscript serve to characterize how the EE, WE, and PE vary in location (latitude) and strength in the different MLT sectors during the different strom phases. Possible relationships to geomagnetic and IMF/SW indices are explored through a regression and correlation analysis.

The results appear reasonable, however, the manuscript presently fails to place the results into the present state of knowledge in the field. One example is that much attention is focused on the latitudinal variability of the EE and WE, where the electrojets are mainly found to move equatorward in the storm main phase, compared to the initial and recovery phase. However, this result in not discussed together with the expanding/contracting nature of the open magnetosphere (e.g. Cowley and Lockwood 1992, Milan 2015), largely explaining why. This needs to be mentioned, and I think the manuscript would benefit from more rather focusing on the specific latitudes that the expansion reach (that they also mention), with relation to e.g. the SYM/H index and the Newell coupling function.

Again, for the discussion section, I would very much appreciate to see how the present results add to the earlier works cited in the introduction. Presently the discussion is largely void of references, and it is not at all clear what the present manuscript has specifically done that is different from the earlier works, other than the method.

Cowley, S. W. H., Lockwood, M. (1992). Excitation and decay of solar wind-driven flows in the magnetosphere-ionophere system. Annales Geophysicae, 10(1–2), 103–115.

Milan, S. E. (2015). Sun et Lumière: Solar Wind-Magnetosphere Coupling as Deduced from Ionospheric Flows and Polar Auroras. In S. Cowley FRS, Stanley W. H. and Southwood, David and Mitton (Ed.), Magnetospheric Plasma Physics: The Impact of Jim Dungey's Research (Vol. 41, pp. 33-64). Springer International Publishing. https://doi.org/10.1007/978-3-319-18359-6

**Specific comments**

The manuscript analyze high latitude ionospheric Hall currents derived from scalar magnetometer meaurements from LEO orbit from the CHAMP satellite during 6 different geomagnetic storm periods. On the basis of these data, the estimated Hall electrojets are characterized in different MLT sectors, and interpreted as either the Eastward (EE), Westward (WE), or Poleward electrojet (PE), the latter occurring only on the dayside in vicinity of the cusp, also in an east/west direction.

**Title:** I find the word "Characteristics" more suitable than "Dynamics" for the title. As the manuscript focus on the correlation of the location and strength of the electrojets deduced from a LEO satellite, with geomagnetic and SW/IMF variables, the study is more characterizing in nature, than describing dynamics, which is usually referred to as changes on shorter time scales than 90 min.

**Here, we completely agree with the Referee and changed the Title accordingly.**

**Abstract:** The extensive listing is unusual. The abstract might be too long for the journal. I would prefer a shorter summary of the method and findings.

**The Abstract is now entirely redrafted, shortened, and we eliminated the extensive listing. We think that it now expresses essentially the main findings, next to a short description of the method.**

**Introduction:** The introduction mention quite a few historic works on the characterization of ionospheric horizontal currents at high latitudes. Although this is highly relevant for the present manuscript, the authors some times fail to convey the implication of the work they describe. The introduction often reads more like a brief summary of the previous works, mainly mentioning what they have been measuring, and not putting it into broader context.

**We have added a few paragraphs to the Introduction, trying to put the topic into broader context. These additions can be found in the first four paragraphs on lines 29-60 together with several new references to historically important publications in our field of research.**

In particular this is evident in the paragraph starting on line 98. I would suggest to remove this paragraph unless the authors can more directly state how this is relevant for the present study.

**This paragraph (previously starting on line 98) has been removed.**

I also have a similar concern with the paragraph starting at line 160. This is mostly a listing of what the referenced literature observe, no explanation of their result is provided. For the reader it would be very satisfactory to briefly explain their findings. Example: Why does the electrojets expand to lower latitudes during increased activity (the oval expands...)? Also, how the electrojets disrupt into space could be elaborated more on, if not removed.

This paragraph has been largely modified and an essential part of it has been shifted to Section 5.2 (lines 513-516) and Section 5.1 (lines 457-475), where it belongs to. As for the latter, some sentences from other places in the former manuscript (e.g., from the former lines 255ff.) have been incorporated there, and we tried to extend the number of relevant references.

The same arguments goes for the paragraph starting at line 176. Please provide how their findings are relevant for the present manuscript. At the moment it only states that Wang et al investigated electrojet currents from space, which is not very informing.

This paragraph (formerly starting at line 176, now at lines 148-158) provides the reader with an outline of the manuscript. The subsequent sections are briefly described and put into a logical order. We think that the paragraph is justified at this place at the end of the Introduction and therefore we kept it as it was.

Possibly you meant the preceding paragraph (formerly starting at line 170), which mention the paper of Wang et al. (2008). This work is quite close to our subject and the authors made use of the same CHAMP data set. They investigated Hall currents for two intense storm periods, but didn't consider the polar electrojet at all. We think that this paper is therefore of relevance for our study and kept this paragraph, though in a slightly modified version (on lines 140-147).

Lines 96-97: "Potential differences at cusp latitudes". In the present manuscript cusp latitudes is refereed to as a point. To get a potential difference, a distance need to be involved. It is not obvious what is the distance in this case. I think you refer to the distance where dayside merging occur. To get around this, maybe an electric field, or velocity value in the east/west direction could be used?

In the present manuscript the cusp is not considered as a point, but as a distance. This distance is determined as the latitudinal interval of the existing polar electrojet (PE, lines 441-445).

Lines 134-159: Suggest to merge this into one paragraph as both paragraphs are dealing with the same Ritter et al 2004 paper.

These two paragraphs have been merged and largely shortened (line 138 ff.). It is also partly shifted to the Method section, where it belongs to (lines 216 ff.).

**Data:** Line 215: Need to define  $\Phi$  (or  $\Phi_{PC}$  as Newell calls it): Magnetic flux inside polar cap. The coupling function describe the change of this related to reconnection at the magnetopause,  $d\Phi_{MP}$ /dt. Together with the nightside reconnection rate ( $d\Phi_N$ /dt), this control the expansion/contraction of the polar cap ( $d\Phi_{PC}$ /dt). This is relevant for the present study as expansion/contractions of the polar cap lead to similar displacements of the electrojets. [Cowlet and Lockwood, 1992].

We reformulated the statements concerning formula (1), aiming at a more precise description as suggested by the Referee, and introduced the term  $\Phi$  (or  $\Phi_{PC}$  as Newell et al, 2007, call it) of open magnetic flux inside the polar cap, as well as the Index N (IndN) as a proxy of  $d\Phi/dt$  (lines 178ff.).

Also relevant for this study is the time averaging of the solar wind data used to estimate IndN. Using a long interval (20-60 min) would likely produce a better correlation to e.g. AL, than using the instant 1 min values. How is this done in the present manuscript?

For the determination of all indices throughout this study, which were available as one-minute cadences, we used time averaging intervals of the overflight intervals (about 20 minutes), as stated at the end of the Data section (lines 194-195).

**Method:** A short introduction to the ionospheric currents at high latitudes would be appreciated, along the lines that we can decompose the currents into Hall, Pedersen, and Birkeland currents, all contributing to the field measured at LEO altitude, but mainly the Hall currents are detected on the ground as explained by Fukushima (1976). As no information regarding the electric field is provided in the manuscript (needed to define the directions of Hall and Pedersen currents), it should state something like "we assume Hall currents to be divergence free, i.e close entirely within the ionosphere, while the Pedersen currents are curl free." Laundal et al. 2018 showed (see their Figure 14) that during summer conditions, which is the case in for the present manuscript, the divergence free and curl free ionospheric currents mainly represented the Hall and Pedersen currents, respectively.

We find this a very good idea and give now at the beginning of this section a short introduction into high latitude currents and the possibilities of their decomposition, as proposed by the Reviewer. These changes can be found in the first three paragraphs of the Method section (lines 197-215).

Lines 241-242: Suggest that you write that only one of the events in Table 1 is presented in the paper, and the rest are included in the Appendix. (a rewrite of the present sentence)

**Done (lines 235-236).**

Line 248: which is the symmetric part of the ring current. As you write on line 252, the ring current is in general asymmetric.

**Corrected (lines 241-243).**

Lines 255-260: I think these sentences belong to the introduction. Is this connection of the EE to the PRC the same as described in lines 166-168? If so, it could be moved into that location to give better background information.

**We shifted this paragraph to a suitable place in Section 5.1 and reformulated it slightly (lines 457ff.), as mentioned already above on the remark about the paragraph that formerly started at line 160.**

Fukushima, N.: Generalized theorem for no ground magnetic effect of vertical currents connected with Pedersen currents in the uniform-conductivity ionosphere, Rept. Ionos. Space Res. Japan, 30, 35–40, 1976

Laundal, K. M., Finlay, C. C., Olsen, N., Reistad, J. P. (2018). Solar wind and seasonal influence on ionospheric currents from Swarm and CHAMP measurements. Journal of Geophysical Research: Space Physics, 123. https://doi.org/10.1029/2018JA025387

**Events:** Line 265: Is this number referring to the SYM/H index? That should be explained before presenting the values. Also a reference to the figure showing SYM/H and the orbits should appear here.

Further, from looking at the SYM/H data at the WDC Kyoto webpage, it is provided as integer numbers. How does the decimal number arise?

The decimal numbers result from averaging the 1-min integer SYM/H values over the interval of observations during the high latitude crossing, as indicated by closely related dashed vertical lines with the abbreviated orbital number in between. To characterize every pass we used peak current location and peak sheet current density only. This is then compared with one mean of any index.

Figure 1 is never introduced. This need to be improved. It would also be of interest to show the AU index in this figure as the paper discusses measurements of the EE during this interval.

Fig. 1 is now introduced in the text (lines 246-251) and, yes, the numbers are referring to the SYM/H index, as it is now explained properly. Further we have added the AU index into the plot as recommended, as well as into the corresponding figures for the other storm intervals in the Appendix / Supplementing material. By the way, we changed the notation from formerly SYM/H and ASYM/H into uniform notations at the Figures and within the text to SymH and AsyH, respectively.

Figure 2: No label or units on the y-axes. Accordingly respectively, and put to the end of the sentence.

The Figure's labeling has been changed, and the caption text corrected, indicating now the unit of the current intensity observations (A/m). This has been done likewise for all the corresponding figures in the Appendix.

Lines 282-286: It is not obvious if the method allows for the determination of the direction of the Hall currents. Is an east/west direction assumed? Sign information is provided in Figure 2, however, the text does not explain what direction positive currents represent (only the figure caption does).

The Hall current intensities, shown in Fig. 2, are now completed with an information about their direction (lines 271-275, as given already in the caption of Fig. 2). And, yes, the single-spacecraft method has to make assumptions about the East-West orientation of (infinite) current sheets, which is now expressed accordingly in the text.

Section 4.1: Although this subsection is devoted to show variations of the observed Hall currents with relation to the SYM/H index, the EE and WE is much discussed. I find it therefore appropriate to mention the expected closer relationship of the observations to the AU and AL index, which is regarded as a measure of the auroral electrojet from the ground.

A corresponding sentence about the AU and AL indices as measures of the electrojet intensities from the ground is now formulated here (lines 278-281).

Lines 378-380: Despite showing the AL index, the authors have not discussed the trend of the measurements with respect to this index.

*Correlations with the ground-based AL index are certainly similar to the AsyH behavior (line 547ff.). This is also seen in Table 3 (page 19).*

Lines380-381: For the reader to judge this statement, a measure of the dependency should be provided, e.g. a correlation coefficient. Although mentioned in the text, this dependency is really hard to see from looking at Figures 1 and 2. This become much more evindent in the later figures where the correlation analysis is performed. This conclusion should therefore not be made at this point.

**This conclusion is kept here, but we added a reference to the Tables with the correlation coefficients later in the manuscript (line 377).**

Section 4.2: From line 363 the manuscript summarizes the observed trends. This should therefore not be a part of subsection 4.2, but rater ha separate subsection (if the summary is really needed), or all the subsections could be combined into one section.

A new subsection 4.3 Summary of the observations has been created (line 358).

**Discussion:** The present discussion does a poor job in relating the observational findings (which is the majority of what is presented in this section) to the state of the art knowledge of the characteristics of the high latitude electrojets. Are the present findings mostly in agreement with the present knowledge, and where does it represent new findings/analysis that could lead to new knowledge? To my knowledge, a possible link between the asymmetric ring current and IMF By has not been established earlier.

That is one piece of information that I think the present manuscript is hinting towards.

Thank you very much for this assessment. We have emphasized this result more clearly in particular within Section 5.1 by introducing two new paragraphs (lines 457-475).

Figure 3: The caption should indicate what the red and blue points refer to.

**Added to the caption of Fig. 3.**

Lines 437-439: If the PE variations during quiet intervals are not due to IMF By, what could it be then? I don't understand this sentence.

**We have reformulated this sentence (lines 532-535).**

Lines 452-489: In my opinion, the term "solar wind electric field penetration" is a misleading term/concept. My take on this is that reconnection opens field-lines on the dayside. During By conditions, these newly opened field lines has a strong bend leading to tension force in the east/west direction. Field-aligned currents facilitate the energy needed to reconfigure these field lines (there will be a drag in the ionsphere due to collissions), and associated east/west plasma flows are seen in the ionosphere, together with the field aligned currents and their associated horizontal currents in the ionosphere, who's magnitude is determined by the ionospheric conductivity. On the ground one observe mainly (at least during summer) the divergence free Hall currents related to this, your PE. In this description it becomes meaningless (in my opinion) to say that the solar wind electric field (originating from the Lorentz transformation) penetrate into the high latitude ionosphere to drive currents. Hence, in order to foster understanding, to distinguish cause and effect, being one of the most

fundamental tasks for physicists, I think the "electric field penetration" is a bad term for this application. That being said, I don't see how these paragraphs really fits into the context of characterizing the PE. I therefore suggest to remove these paragraphs.

**Yes, you are right, the term "solar wind electric field penetration" and the physical concept behind it is misleading. We omitted these paragraphs completely in the revised manuscript.**

Lines 495-499: This result has the strongest correlation. Could the authors try to elaborate on possible explanations for this result? The ASYM/H describe to some extent an asymmetry in the inner magnetosphere. The PE arises, as the manuscript shows, due to IMF By, which is maybe the most important paramater affecting the asymmetry in the magnetosphere. From simple linear correlation of IMF Byl and ASYM/H one get r=0.29, while with SYM/H one get r=-0.16 (using 5 min OMNI data from 1981-2016). This supports the results in the manuscript, and suggests that IMF By could be a source of asymmetry in the ring current.

**As stated before, we added two paragraphs (lines 457-475), which try to give physical explanations of this result. A part of this text was originally placed within the 'Method' section.**

Line 540: What is meant by "changing SYM/H" index? Is there a criterion in the data selection that the

SYM/H should drop by e.g. a fixed amount? If so, how what is the reason for that?

This sentence is corrected to "...show cases of AE appearance with a change of the SYM/H index" (line 535). No, there is no such criterion; the former sentence was capable for being misunderstood. Hopefully it is more clear now.

Line 551: Strictly speaking, IndN does not characterize a current function, but is designed to be proportional to the amount of opened magnetic flux per unit time, which in turn affect the current systems.

**The sentence in question is now reformulated to address cause and effect more clearly (lines 544-546).**

Line 563: Need to introduce Figure 5.

Introduced now (line 559).

Line 282-583: The mentioned trend of Mlat vs IndN is not obvious from Figure 6d.

This is reformulated now with more cautiousness (lines 731-733).

Lines 595-596: I disagree. As they should in principle be the same, the large observed spread (Figure 7c) give insight to the difference in technique used to probe the WE (CHAMP and AL index). I was surprised the relationship was not clearer. Could some of this variation be due to difference is in sampling/averaging window?

The Discussion section, now entitled "Correlation analyses and discussion", has largely been redesigned and reformulated, in particular with respect to Section 5.1 on the Polar Electrojet.

**Conclusion:** Similar to my comment regarding the abstract, I find the conclusion section very long. It is a very comprehensive listing of the detailed discussion provided earlier. I would have appreciated if the authors would have rather summarized what they believe is the most important and significant among their findings.

The Conclusions have been radically shorten and focused on the main findings.

Lines 606-712: This has not been mentioned before and should therefore be moved to the discussion

This long part of the former manuscript has been shifted from the Conclusions to the Discussion section, while the Conclusions section has been completely redrafted and considerably shortened. The former subsections about Hall current behavior form now a subsection within the Discussion.

**Minor comments**

We made all the minor corrections/changes according to the Referee's recommendations. Thank you for pinpointing all these details.

Line 10: IndN needs to be defines. Suggest that you only refer to it as the Newell et al coupling function in the abstract.

Line 24: Should this be 80 degrees MLAT, to be consistent with the equatorward motion mentioned?

Line 28: Need to define acronyms in abstract. Or maybe just refer to it as ring current in the abstract.

Lines 74-77: What level of confidence does this refer to? Could a reference or a name of the formula be provided?

Line 111: compilation comparison

Line 112: demonstrates

Line 131-132: From looking at Figure 26 in Sandholt et al 2004, theese fast convection channels are located on open field lines, hence poleward of the OCB and not at the OCB as the manuscript states.

Lines 134, 139, and 153, and throughout the text: The use of "density" refering to the strength of the electrojets is sometimes unfortunate in my opinion. It can easily me mixed with how dense the current lines in the model is placed. Consider to change it to "strength".

We changed this term "density" throughout the manuscript to either "strength" or "intensity" in accordance with the respective context.

Line 151: Delete "currents"

Line 160: EISCAT is an incoherent scatter radar network, not a magnetomenter network

*"EISCAT" was till 1993 the name of a magnetometer chain (mentioned on line 514); the new name "IMAGE" was received only in 1997.*

Line 180: electromagnetic

Line 207: 12 observatories

Line 219: BT is actually only constricted from the y and z components of IMF

Line 220: One should rather use theta = arctan2(By,Bz), which is a common algorithm that handle the different signs of By and Bz properly, compared to the mathematical function arctan().

Line 233: delete "there"

Line 236: density level

Line 245: Suggest to delete "mean"

Line 282: density intensity

Line 331: delete "there"

Line 395: Transferred is not a good word here.

Lines 405-406: Suggest to replace "occurrence and dynamics" with "strength" and "domains" with "processes"

Line 409: Delete "from geomagnetic activity researches"

Lines 411-412: This sentece is not related to any of the following text.

Line 415: add "all the" before 6

Line 414: Add something like "versus IMF and geomagnetic indices"

Lines 419-420: I dont understand this sentence.

Line 511: Suggest to rewrite: "... correlation of IndN with PE Mlat (r=0.52)."

Lines 517-518: Suggest to delete this sentence.

Line 528: Suggest to add something like "from the individual orbits". As I read these figures, each dot correspond to a specific orbit within the MLT range, among the 6 events studied.

**Anonymous Referee #2 : angeo-2018-31**

"Dynamics of the electrojet during intense magnetic disturbances" by Liudmila I. Gromova et al. Interactive comment on Ann. Geophys. Discuss., https://doi.org/10.5194/angeo-2018-31, 2018

We would also like to thank the reviewer #2 for the thorough evaluation of our paper. Major items of this report are conform to the critics of Referee #1. We would therefore ask this Referee to consider also our answers to Referee #1. Alike there, we have answered all comments by inserting our response behind the comment in italic and blue. Further, we include the revised manuscript with corrections indicated in red for the Referee's convenience.

In the manuscript the authors analyze total magnetic field data from the CHAMP satellite during six periods of magnetic storms with respect to the relation between the phases of the storms and those of the polar and auroral electrojets. A somewhat complex procedure allows deriving sheet current densities of the Hall currents flowing underneath the spacecraft. The measurements were made during Northern summer periods and stem from noon-midnight as well as dawn-dusk crossings of the spacecraft. The main products consist of correlations between the measured Hall current locations and strengths with auroral and ring current activity as quantified by various geomagnetic indices. The results allow some interesting insights into the mutual dependences or lack of them.

The presented text contains some valuable data deserving publication. However, the presentation is too detailed and tedious to read. The main products are well presented in Figures 4 to 7 and their discussion appropriate. However, the conclusions are not concise, but rather repetitive of the earlier description of the data. One misses interpretations in terms of the build-up of the asymmetric and symmetric ring currents and the role played in that by the high-latitude magnetosphere as indicated by the behavior of the polar and auroral electrojets. I recommend to encourage the authors to shorten strongly Abstract, Introduction, and Conclusions and concentrate in the latter on the meaning of the most striking correlations.

The Abstract and the Conclusions are now entirely redrafted, shortened, and we eliminated the extensive listings. We think that they now express essentially the main findings of the study.

The Introduction has been extended by more references to important ideas and science developments in this field of research and we tried to place our study appropriately into the present state of knowledge, in particular by discussing the expanding/contracting nature of the open magnetosphere. Two additional comments.

The polar and auroral electrojets are treated as if there was no relation. One is missing a reference to the Region 1 and 2 field-aligned current systems.

**Region 1 and 2 FACs (or Birkeland currents) are now mentioned in the Introduction and the Discussion makes reference to these terms as well.**

Pages 17 and 18 contain estimates of electric field penetration from the solar wind into the magnetosphere. The authors seem to take this concept literally. There is no direct electric field penetration. There is reconnection and maybe some viscous interaction by plasma waves between the solar wind and the Earth's magnetic field. Thereby forces are being transferred via magnetic shear stresses. The direction of By matters of course. Electric fields inside the magnetosphere stem from the application of these stresses to the ionospheric plasma. Therefore, the calculated percentages of electric field transfer from a partial magnetic field component are meaningless.

Yes, you are right, the term "solar wind electric field penetration" and the physical concept behind it is misleading. We omitted these paragraphs completely in the revised manuscript.

[revised manuscript text omitted]

---

## Referee Report (RR1)

Here follow my comments to the response to my review of the manuscript "Characteristics of the electrojet during intense magnetic disturbances" by Gromova et al. All line number refer to the tracked changes version of the revised manuscript unless other is specified.

Abstract:
Line 47: "Substorms occurring during daytime before the storm main phase (…) ": The substorm is mainly a nightside phenomena. Not sure what is meant here. Suggest to delete "during daytime".

Introduction:
Line 222: Add: "They found the intensity ..." at the beginning of the sentence

Data:
Line 277-278: Here, the revised manuscript state that the indices has been averaged over the overflight intervals. However, in the response letter, a one minute resolution is stated in the response to my comment regarding this. I suspect that the revised manuscript is correct and the response letter is wrong. Please clarify.

Method:
Line 299: "The hall current at high latitudes (…)" (delete flowing)

Section 4:
The response letter claim to have added the sentence "The correlations with the ground based AL index are similar to the ASYM/H behavior." However, this referee cannot find this sentence in the revised manuscript. Have the authors forgotten to place this sentence in the revised manuscript?

Regarding the next comment in my initial response, referring to line 380-381 in the initial submission: A reference to the relevant Table is not included in the revised manuscript at this specific location, lines 475-476.  As the reader can hardly see this trend from Figures 1 and 2 alone, this conclusion should either follow the subsequent analysis shown, or it should be stated here that further analysis shows this trend (the correlation with AsymH). Otherwise, this conclusion will not be sufficiently supported at this point.

Discussion:
Table 2 caption: It should be stated here that these results are for the PE current system.

Regarding the new material on the correlation between IMF By and AsymH, lines 601-619: I have a hard time understanding any direct link between the PRC and the PE currents. On the dayside, the PE are mainly on open field-lines,  close to the OCB, at 75-80 degrees MLAT as Figure 3 shows. This is different from the EE location (Figures 4-5). The PRC is in the inner magnetosphere, on closed field-lines, and is traditionally believed to closed somewhere in the dusk sector in the region of EE (as mentioned in the previous version of the manuscript). I would encourage the authors to relax the language when suggesting this as a possible link, as this has hot been established earlier. It need to be specified that this is a speculation and that the observed correlation also can have other explanations, rather than stating that this is *the* explanation.

Line 747-748: The authors has not provided any relevant response this point (last point from the discussion section in my first review). I still think the AL vs WE  comparison still deserve some attention in the manuscript.

Conclusions:
Line 782: Should "displaces" be "is located"?

From the list of minor comments from my first review, I was not able to find the response to the following: Lines 120-122: What level of confidence does this refer to? Could a reference or a name of the formula be provided?

---

## Referee Report (RR2)

Review of the revised Manuscript angeo-2018-31: "Characteristics of the electrojet during intense magnetic disturbances" by L. I. Gromova et al.

In the manuscript the authors analyze total magnetic field data from the CHAMP satellite during six periods of magnetic storms with respect to the relation between the phases of the storms and those of the polar and auroral electrojets. A somewhat complex procedure allows deriving sheet current densities of the Hall currents flowing underneath the spacecraft. The measurements were made during Northern summer periods and stem from noon-midnight as well as dawn-dusk crossings of the spacecraft. The main products consist of correlations between the measured Hall current locations and strengths with auroral and ring current activity as quantified by various geomagnetic indices. The results allow some interesting insights into the mutual dependences or lack of them.

The authors have responded very well to the extensive comments of Reviewer #1 and also to the more general ones which I raised. Abstract, Introduction, and Conclusions have improved and the referenced literature has been extended. I have no new reservations and recommend the paper for publication.

---

## Author Response (AR2)

**Anonymous Referee #1 : angeo-2018-31, revised version**

**"Characteristics of the electrojet during intense magnetic disturbances"  by  Liudmila I. Gromova et al.**
* * *
*We would like to thank the Reviewers again for the thorough evaluation of our paper. We appreciate their constructive comments and criticism. As before, we have answered all comments by inserting our response behind the comment in italic and blue. For convenience, we include again the corrected manuscript version with changes indicated in red. The line numbers below refer to this version.*

Here follow my comments to the response to my review of the manuscript "Characteristics of the electrojet during intense magnetic disturbances" by Gromova et al. All line number refer to the tracked changes version of the revised manuscript unless other is specified.

Abstract:

Line 47: "Substorms occurring during daytime before the storm main phase (...) ": The substorm is mainly a nightside phenomena. Not sure what is meant here. Suggest to delete "during daytime".

*D'accord; we deleted "during daytime" as suggested (line 19).*

Introduction:

Line 222: Add: "They found the intensity ..." at the beginning of the sentence

*We added "They found ..." at the beginning of this sentence (line 146).*

Data:

Line 277-278: Here, the revised manuscript state that the indices has been averaged over the overflight intervals. However, in the response letter, a one minute resolution is stated in the response to my comment regarding this. I suspect that the revised manuscript is correct and the response letter is wrong. Please clarify.

*Referee #1 obviously refers to the first variant of our replies to the Referees, dated Aug 21, 2018. In the revised version of our responses of Sep 06, 2018, we had already corrected this, stating that "For the determination of all indices throughout this study, which were available as one-minute cadences, we used time averaging intervals of the overflight intervals (about 20 minutes)..." (lines 194-195).*

Method:

Line 299: "The hall current at high latitudes (...)" (delete flowing)

*Deleted (line 216).*

Section 4:

The response letter claim to have added the sentence "The correlations with the ground based AL index are similar to the ASYM/H behavior." However, this referee cannot find this sentence in the revised manuscript. Have the authors forgotten to place this sentence in the revised manuscript? Regarding the next comment in my initial response, referring to line 380-381 in the initial submission: A reference to the relevant Table is not included in the revised manuscript at this specific location, lines 475-476. As the reader can hardly see this trend from Figures 1 and 2 alone, this conclusion should either follow the subsequent analysis shown, or it should be stated here that further analysis shows this trend (the correlation with AsymH). Otherwise, this conclusion will not be sufficiently supported at this point.

*Again, Referee #1 refers here to the first variant of our reply, while in the second, final variant we didn't make such a statement, because this sentence was eliminated in the final manuscript. However, reconsidering this subject according to the Referee's comment, we included this statement now within section 5.2.1 (lines 549-550).*

Discussion:

Table 2 caption: It should be stated here that these results are for the PE current system.

*We added this sentence at the beginning of the caption of Table 2: "Correlations of the PE current system with various indices. The columns show the dependent..." (line 416).*

Regarding the new material on the correlation between IMF By and AsymH, lines 601-619: I have a hard time understanding any direct link between the PRC and the PE currents. On the dayside, the PE are mainly on open field-lines, close to the OCB, at 75-80 degrees MLAT as Figure 3 shows. This is different from the EE location (Figures 4-5). The PRC is in the inner magnetosphere, on closed field-lines, and is traditionally believed to closed somewhere in the dusk sector in the region of EE (as mentioned in the previous version of the manuscript). I would encourage the authors to relax the language when suggesting this as a possible link, as this has hot been established earlier. It need to be specified that this is a speculation and that the observed correlation also can have other explanations, rather than stating that this is the explanation.

*We tried to give a "more relaxed" statement and pinpoint to the possibility of other explanations, as suggested by the Referee (lines 474-476).*

Line 747-748: The authors has not provided any relevant response this point (last point from the discussion section in my first review). I still think the AL vs WE comparison still deserve some attention in the manuscript.

*We added the following sentences with regard to AL and WE comparison (lines 600 ff.): "The AL index characterizes likewise the westward electrojet WE, but the method of its derivation differs. The AL index indicates the maximum decrease in the horizontal magnetic component of a longitudinal chain of observatories, which is equivalent to the current intensity in westward direction for a given UT moment. The WE, on the other hand, marks the maximum current intensity in westward direction*

*over the meridional sector and the time interval of the spacecraft's orbit. The differences in the methodology of their determination results in rather low correlation values and in the MLT dependent variations of the correlation."*

Conclusions:

Line 782: Should "displaces" be "is located"?

*Corrected (line 638).*

From the list of minor comments from my first review, I was not able to find the response to the following: Lines 120-122: What level of confidence does this refer to? Could a reference or a name of the formula be provided?

*We added the phrase in parentheses (line 87): "(corresponding to 95% confidence interval)", while refraining from an additional reference, because this matter can be found in statistics textbooks like, e.g., the book on "Statistische Physik in Beispielen" by H. Schilling, Fachbuchverlag Leipzig, 1972.*

[revised manuscript text omitted]